# Causal Mixture Models:
# Characterization and Discovery

**Sarah Mameche**
CISPA Helmholtz Center
for Information Security
sarah.mameche@cispa.de

**Janis Kalofolias**
CISPA Helmholtz Center
for Information Security
janis.kalofolias@cispa.de

**Jilles Vreeken**
CISPA Helmholtz Center
for Information Security
jv@cispa.de

## Abstract

Real-world datasets are often a combination of unobserved subpopulations that follow distinct causal generating processes. In an observational study, for example, participants may fall into unknown groups that either (a) respond effectively to a drug, or (b) show no response due to drug resistance. Not accounting for such heterogeneity then risks biased estimates of drug effectiveness. In this work, we formulate this setting through a causal mixture model, in which the data-generating process of each variable depends on latent group membership (a or b). Specifically, we model each variable as a mixture of structural causal equation models, where latent categorical (mixing) variables index assignment to subpopulations. Unlike prior work, the approach allows for multiple independent mixing variables, each affecting distinct sets of observed variables. To infer both the graph, mixing variables, and assignments jointly, we integrate mixture modeling into score-based causal discovery; show theoretically that the resulting scoring criterion is consistent; and demonstrate empirically that the resulting causal discovery approach discovers the causal model in synthetic and real-world evaluations.

## 1 Introduction

A central part of scientific investigation is understanding cause-effect relationships, which the field of causal discovery [Pearl, 2009] aims to discover directly from observational data. Many existing causal discovery approaches, however, rely on idealized assumptions, among others assuming that no relevant unmeasured variables exist and that all samples come from a homogeneous distribution. Real-world applications might violate both assumptions, for example, when observations come from heterogeneous populations or environments.

Take, for example, a nationwide study of antimicrobial resistance in hospitalized patients, focusing on a resistant pathogen such as Methicillin-Resistant Staphylococcus Aureus (MRSA) [Hasanpour et al., 2023]. As patients come from different regions, their individual medical histories differ, including prior exposure to pathogens such as Enterococcus [Li et al., 2022] with known cross-resistance to MRSA. The regional plasmid profiles of Enterococcus largely determine its susceptibility, say to Vancomycin [Boumasmoud et al., 2022], in turn influencing MRSA cross-resistance [Arredondo-Alonso et al., 2020]. Although well documented, this variable is not routinely measured, and is hence a latent variable that defines the mechanism under which the presence of Enteroccocus affects MRSA cross-resistance [Cong et al., 2019]. Consequently, observations across regions effectively arise from a mixture of distinct causal mechanisms.

More generally, observational data can be a combination of multiple subgroups with distinct generating processes. To illustrate, consider a simplified, synthetic setting in Fig. 1, where the causal mechanism for $Y$ is a mixture of two functional relationships $X \to Y$. Treating all samples as one population can then cause artifacts during causal discovery, such as spurious relationships or reversed

39th Conference on Neural Information Processing Systems (NeurIPS 2025).

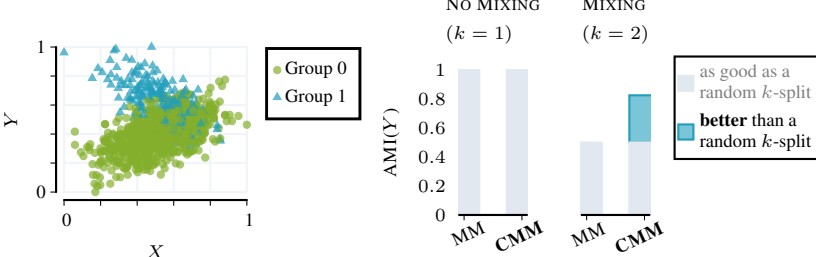

Figure 1: *Causal Mixture Models.* Left: Example of a mixed causal relationship $X \to Y$. Right: Recovering the class assignments with (Gaussian) mixture models (MM) compared to causal mixture models (CMM) on simulated bivariate datasets.

causation [Huang et al., 2020]. While a line of research known as multi-context and interventional causal discovery [Mooij et al., 2016; Huang et al., 2020; Squires et al., 2020] addresses a similar setting, these require access to multiple datasets where all causal mechanisms remain fixed, which are unknown in the motivating example.

Recent works attempt to separate a single given dataset into multiple interventional datasets or environments; for instance, Kumar et al., 2024 combine Gaussian mixture modeling (GMM) with interventional discovery (UT-IGSP), addressing . mixtures arising from atomic interventions [Kumar et al., 2024]. In examples such as Fig. 1, the underlying mixtures are better captured through conditional relationships, such as in mixture-of-regressions models [Hennig, 2000]. To illustrate the difference between an unconditional mixture model (MM) and the conditional (causal) counterpart (CMM), we draw synthetic datasets over $X, Y$ and evaluate the discovered class assignments for $Y$ via the Adjusted Mutual Information (AMI). While in the case without mixing ($k = 1$ group), both methods correctly reject the existence of groups, with mixing ($k = 2$ groups), the MM reports no better than random assignments, whereas the CMM improves over the baseline. This suggests considering the causal mechanism for $Y$ as a conditional mixture subject to a latent variable $Z$.

When extending the scope from the bivariate case to multiple observed variables, our interest then also extends from a single to multiple, independent latent variables. Returning to the motivating example, for instance, suppose each medical center chooses a different collaborating laboratory for antibiotic testing. This introduces a batch effect, a latent factor independent of regional susceptibility. To address this, different from previous formalizations that assume a single mixing variable affecting all observables [Mazaheri et al., 2023; Kumar et al., 2024], we allow multiple independent mixing variables that each affect different subsets of observed variables (cf. Fig. 2). For each variable, we model its generating mechanism as a conditional mixture given its direct causes and its corresponding latent factor, referred to as a Causal Mixture Model (CMM). The central question in the remainder of this work is how to infer the structure of such models from data.

**Causal Mixture Models**   To summarize, we propose basing mixture modeling on functional relationships within a causal graph, where we consider the causal mechanism for each variable as a mixture of conditional relationships given its causes and an associated latent variable. Unlike two-stage approaches that separate cluster and graph discovery, our approach integrates mixture inference directly into causal discovery by extending local score-based criteria, where we focus on linear mixture-of-regression (MLR) models inferred through Expectation Maximization (EM) and extend the BIC score [Chickering, 2002]. We show that under oracle access to the MLR parameters, we can guarantee the identification of the causal model under mild assumptions, and hypothesize that this also holds for their EM estimates in practice. We propose integrating this approach into score-based causal discovery algorithms such as Greedy Equivalence Search (GES) [Chickering, 2002] or TOPIC [Xu et al., 2025]. To demonstrate empirically we can recover the CMM components in practice, we consider simulated mixed data, a mixture of interventions [Kumar et al., 2024], as well as a real-world benchmark on causal cell signaling pathways [Sachs et al., 2005].

We include all theoretical justifications and experimental details in the supplement.

## 2 Causal Model

Given a set of continuous random variables $\boldsymbol{X} = \{X_1, \ldots, X_n\}$, we are interested not only in causal relationships among them, but especially in unknown changes of their causal mechanisms. Assuming that the mechanisms are linear functions of a fixed subset of the observed random variables, we allow their coefficients to be chosen from a finite set of vectors conditional on an external latent variable.

To formalise this, we also consider a set of discrete, unobserved random variables $\boldsymbol{Z} = \{Z_1, \ldots, Z_m\}$, with $m \leq n$, each following a categorical distribution $Z_i \sim \text{Categorical}(\boldsymbol{\gamma}^i)$ with $Z_i \in \{1, \ldots, K_i\}$. That is, each $\boldsymbol{\gamma}^i$ lies on a $K_i$-dimensional probability simplex $\boldsymbol{\gamma}^i = (\gamma_1^i, \ldots, \gamma_{K_i}^i)$ with $\sum_{k=1}^{K_i} \gamma_k^i = 1$, so that $\mathbb{P}(Z_i = k) = \gamma_k^i$. We call these random variables *mixing variables*.

The causal mechanism of each observed random variable in $\boldsymbol{X}$ depends, besides on the set of observed causal parents denoted $\mathbf{Pa}_j$, on exactly one of the unobserved, latent $\boldsymbol{Z}$, as determined by the surjective map $\text{La} : \boldsymbol{X} \to \boldsymbol{Z} : X_j \mapsto Z_i$, in which case we simply write $\text{La}_j = Z_i$. The mixing variable directly affects the parameters of the causal mechanism, which we therefore express as a function $b_j : [K_i] \to \mathbb{R}^{|\mathbf{Pa}_j|} : z \mapsto \beta_{jz}$ mapping each value $z$ of $Z_i$ to a linear coefficient vector $\beta_{jz} \in \mathbb{R}^{|\mathbf{Pa}_j|}$. Hence, the parameters of the functional dependency $f$ is the collection of vectors $B_j = (\beta_{j1}, \ldots, \beta_{jK_i})$; this consists of one coefficient vector $\beta_{jk}$ for each mixing coefficient $1 \leq k \leq K_i$, and each such vector has dimension equal to the number of parents $\mathbf{Pa}_j$ of $X_j$.

Summarizing, we can now model each random variable $X_j$ as generated from its observed causes $\mathbf{Pa}_j \subseteq \boldsymbol{X}$ by the causal function $f$ and the coefficients $b_j$ that depend on the latent $Z_i = \text{La}_j \in \boldsymbol{Z}$, where we recall that $\boldsymbol{Z} \cap \boldsymbol{X} = \emptyset$. Then, we have

$$X_j = f(\mathbf{Pa}_j, b_j) + N_j \qquad \text{with} \qquad f(x, b_j(z)) = \beta_{jz}^\top x + \beta_{jz}^{(0)}, \tag{1}$$

where $N_j \perp\!\!\!\perp \mathbf{Pa}_j$ is additive Gaussian noise, $N_j \sim \mathcal{N}(0, \sigma^2)$.

This construction implies that for a random variable $X_j$ for which $\text{La}_j = Z_i$ we get

$$(X_j | \mathbf{Pa}_j = \mathbf{y}, \text{La}_j = k) \sim \mathcal{N}\left(\boldsymbol{\beta}_{jk}^\top \mathbf{y}, \sigma^2\right) \qquad \text{for } k \in \{1, \ldots, K_i\} \text{ and} \tag{2}$$

$$(X_j | \mathbf{Pa}_j = \mathbf{y}) \qquad \sim \text{MLR}\left(\mathbf{B}_j, \boldsymbol{\gamma}^j, \sigma^2\right), \tag{3}$$

where $\text{MLR}\left(\mathbf{B}, \boldsymbol{\gamma}, \sigma^2\right)$ is the conditional distribution of a mixture of linear regressions with density

$$p_{X|\mathbf{Y}}^{\text{MLR}}(x, \mathbf{y}; \mathbf{B}, \boldsymbol{\gamma}, \sigma^2) = \sum_{k=1}^K \gamma_k p_X^{\mathcal{N}}(x; \boldsymbol{\beta}_k^\top \mathbf{y}, \sigma^2) = \sum_{k=1}^K \frac{\gamma_k}{\sqrt{2\pi}\sigma} \exp\left(-\frac{\|\boldsymbol{\beta}_k^\top \mathbf{y} - x\|^2}{2\sigma^2}\right), \tag{4}$$

and $p_X^{\mathcal{N}}(x; \mu, \sigma^2)$ is the density of the normal distribution with mean $\mu$ and variance $\sigma^2$. Note that in case $\mathbf{Pa}_j = \emptyset$, the formulation reduces to a standard unconditional GMM.

**Graphical Model**  We represent the above causal model as a Directed Acyclic Graph (DAG) $\mathcal{G}^Z = (\boldsymbol{X} \cup \boldsymbol{Z}, E^Z)$ over both observed $\boldsymbol{X}$ and latent $\boldsymbol{Z}$ random variables, between which we add edges $X_j \to X_l$ whenever $X_j$ is a cause of $X_l$, as well as $Z_i \to X_j$ whenever $\text{La}_j = Z_i$; since the $\boldsymbol{Z}$ were assumed exogenous and independent, we allow no incoming edges toward the $\boldsymbol{Z}$ themselves. Of special interest is the subgraph $\mathcal{G} = (\boldsymbol{X}, E)$ that is induced on $\mathcal{G}^Z$ by the node subset $\boldsymbol{X}$, with $E = \{X_j \to X_l \mid X_j \to X_l \in E^Z\}$. Considering only the latter subgraph, we denote the set of all observed direct predecessors of $X_j$ in $\mathcal{G}$ by $\mathbf{Pa}_j \subset \boldsymbol{X} \setminus \{X_j\}$. We also consider a surjective map $\pi : \boldsymbol{X} \to \mathbb{N}$ which induces a topological ordering on $\mathcal{G}$ (and thus partial ordering over $\boldsymbol{X}$) that we call a *causal order* under $\mathcal{G}$, and assigns such values on $\boldsymbol{X}$ that $X_r \in \mathbf{Pre}_j \Rightarrow \pi(j) < \pi(r)$, for $\mathbf{Pre}_j \subset \boldsymbol{X} \setminus \{X_j\}$ the set of all

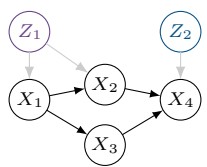

Figure 2: *Example Causal Mixture Model. The observed variables are affected by two latent mixing variables $Z_1, Z_2$.*

direct and indirect predecessors of $X_j$ in $\mathcal{G}$. Fig. 2 depicts an example of such a graph $\mathcal{G}^Z$ (colored and black) and the resp. induced subgraph $\mathcal{G}$ (black). We then model edges such as $Z_1 \to X_1$ where $X_1$ is a source node of the causal graph as a GMM and edges such as $Z_1 \to X_2$ through an MLR.

We assume the causal Markov Property to hold in the full causal model $\mathcal{G}^Z$, which results in the following factorization over the observed variables.

**Assumption 1** (Mixture-Markov Property). The distribution of $\boldsymbol{X}$ has (marginal) density

$$p_{\boldsymbol{X}}(\mathbf{x}) = \prod_{X_j \in \boldsymbol{X}} p_{X|\mathbf{Pa}_j}^{\mathrm{MLR}}(x, \mathbf{pa}_j; \mathbf{B}, \boldsymbol{\gamma}, \sigma^2) , \tag{5}$$

where we implicitly use the independence of all $\boldsymbol{Z}$ (see Appendix A.3 on relaxing this assumption).

The objective in this work is to infer the CMM from finite observations, as follows.

**Problem 1.** *Given i.i.d. observations $\mathcal{D} = \{\mathbf{x}_1, \ldots, \mathbf{x}_r\}$, of the random variables $\boldsymbol{X}$ generated from a distribution compatible with our assumptions, we aim to infer the structure of $\mathcal{G}^Z$,*

1. *the causal dependencies $X_j \rightarrow X_l \in E$ encoded in the edges $E$ of the induced graph $\mathcal{G}$,*
2. *the linear coefficients of the corresponding causal mechanisms $\boldsymbol{\gamma}^j$ for all $X_j \in \boldsymbol{X}$,*
3. *the set $\boldsymbol{Z}$ of latent variables, i.e., their number $m$, and for each $Z_i \in \boldsymbol{Z}$ the domain $K_i$ and mixing probabilities $\boldsymbol{\gamma}^i$, as well as*
4. *the mapping $\mathrm{La}_j \in \boldsymbol{Z}$ for each $X_j \in \boldsymbol{X}$.*

We address this problem in the following section.

## 3 Theory

As the first contribution of our work, we study conditions that allow inference on $\mathcal{G}^Z$, as formalised in Problem 1, using score-based causal discovery. To this end, it is of interest to determine whether existing local scoring criteria can be adapted for our setting, so as to use them consistently in a score-based framework, such as the well-known Greedy Equivalent Search (GES) algorithm [Chickering, 2002] and related approaches [Xu et al., 2025].

GES performs a greedy search over the set of all Markov equivalent classes (MECs) of causal graphs with nodes $\boldsymbol{X}$. More specifically, each iteration updates the current MEC by the best hypothesis among all those that differ from the current one by a single edge modification (i.e., edge reversal, addition, or removal). This process allows for the inference of the most likely MEC, as expressed in the form of the edges of a Partially Directed Acyclic Graph (PDAG) with nodes representing the observed $\boldsymbol{X}$.

Importantly, GES can be shown to be asymptotically consistent as long as the search is guided by a scoring criterion that satisfies the appropriate criteria given shortly. We write $\mathcal{L}_{\boldsymbol{X}} \in I_{\mathcal{G}}$ for a graph with nodes $\boldsymbol{X}$ whenever the (conditional) independencies implied by the PDAG $\mathcal{G}$ are also true for the distribution $\mathcal{L}_{\boldsymbol{X}}$.

**Definition 3.1** (Consistent Scoring Criterion [Chickering, 2002]). Given data $\mathcal{D}$ of size $r$ sampled from distribution $\mathcal{L}_{\boldsymbol{X}}$ and two graph hypotheses $\mathcal{G}_1^{\mathrm{h}}, \mathcal{G}_2^{\mathrm{h}}$ then the score $S$ is consistent whenever

1. $\mathcal{L}_{\boldsymbol{X}} \in I_{\mathcal{G}_1^{\mathrm{h}}}$ and $\mathcal{L}_{\boldsymbol{X}} \notin I_{\mathcal{G}_2^{\mathrm{h}}} \implies S(\mathcal{G}_1^{\mathrm{h}}; \mathcal{D}) > S(\mathcal{G}_2^{\mathrm{h}}; \mathcal{D})$
2. $\mathcal{L}_{\boldsymbol{X}} \in I_{\mathcal{G}_1^{\mathrm{h}}} \cap I_{\mathcal{G}_2^{\mathrm{h}}}$ and $\mathcal{G}_1^{\mathrm{h}}$ has fewer parameters than $\mathcal{G}_2^{\mathrm{h}}$ it is $S(\mathcal{G}_1^{\mathrm{h}}; \mathcal{D}) > S(\mathcal{G}_2^{\mathrm{h}}; \mathcal{D})$.

One score that satisfies this criterion is the Bayes Information Criterion (BIC).

**Definition 3.2** (Bayes Information Criterion). Given model hypothesis $\mathcal{H}$ assuming distribution $\mathcal{L}_{\boldsymbol{X}}(\boldsymbol{\theta})$ with parameters $\boldsymbol{\theta} \in \Theta \subseteq \mathbb{R}^d$ and observations $\mathcal{D} = \{\mathbf{x}_1, \ldots, \mathbf{x}_r\}$ of $\boldsymbol{X}$, the score of $\mathcal{H}$ is

$$\mathrm{BIC}(\mathcal{H}) \coloneqq -2 \log p_{\boldsymbol{X}}(\mathcal{D}|\hat{\boldsymbol{\theta}}) + d \log r , \tag{6}$$

where the first term is the scaled (log) likelihood of $\mathcal{L}_{\boldsymbol{X}}$ evaluated at its maximiser $\hat{\boldsymbol{\theta}}$.

This score has the important property of decomposability [Chickering, 2002], dictating that when $\mathcal{L}_{\boldsymbol{X}}$ factorises as a product the BIC decomposes as a sum

$$p_{\boldsymbol{X}}(\mathbf{x}) = \prod_{X_j \in \boldsymbol{X}} p_{X_j|\mathbf{Pa}_j}(x_j|\mathbf{pa}_j) \implies \mathrm{BIC}(\boldsymbol{X}) = \sum_{X_j \in \boldsymbol{X}} \mathrm{BIC}(X_j|\mathbf{Pa}_j) . \tag{7}$$

To transfer these properties to our own model, we need to further account for the latent mixing variables. Hence, at each step of the GES algorithm, and according to our model of Eq. (5) we need to compute the likelihood $p_{X|\mathbf{Pa}_j}^{\mathrm{MLR}}(x, \mathbf{pa}_j; \mathbf{B}, \boldsymbol{\gamma}, \sigma^2)$, which requires an estimation of the Mixture of

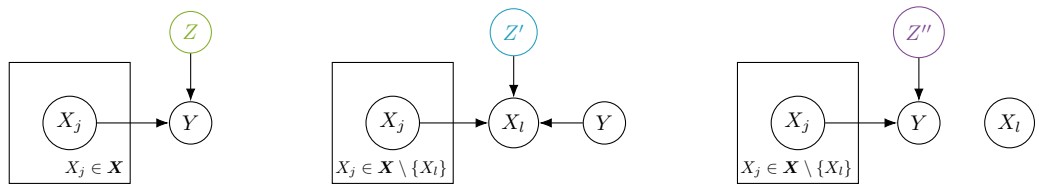

Figure 3: *Identifiable Cases.* Under mild assumptions, when $Z$ has at least two mixing components, all of the shown DAGs are identifiable.

Linear Regressions (MLR) parameters. Unfortunately, the computation of the Maximum Likelihood Estimate (MLE) estimates for both Gaussian Mixture Models (GMMs) and MLR models is NP-hard in the general case. However, an efficient algorithm to compute well-performing local maximisers of the corresponding likelihoods is the Expectation Maximisation (EM) framework [Wu, 1983] and its variants [Bishop, 2006]; we therefore employ these algorithms to approximate the MLE parameters.

Although certain conservative claims can be made for the consistency of EM, we postpone them for later on. Instead, for the first part of our analysis we assume the availability of an oracle that can compute the maximum likelihood estimates for the MLR model for a given number of mixtures $K$,

$$\hat{\boldsymbol{\theta}} = (\hat{\mathbf{B}}, \hat{\boldsymbol{\gamma}}, \hat{\sigma}) = \underset{(\mathbf{B}, \boldsymbol{\gamma}, \sigma) \in \Theta_K}{\arg\max} \; p_{X|Y}^{\text{MLR}}(x, \mathbf{y}; \mathbf{B}, \boldsymbol{\gamma}, \sigma^2), \tag{8}$$

where $\Theta_K = (\mathbb{R}^{|\mathbf{Pa}_j|})^K \times \mathcal{S}^{K-1} \times \mathbb{R}_+$ is the space of the model parameters for $K$ mixtures and $\mathcal{S}^K$ the $K$-dimensional probability simplex. Given the availability of such an oracle, we can adapt the GES algorithm in a way that both preserves its asymptotic consistency and takes into account any possible latent variables. For this, we also incorporate the asymptotically consistent estimation of the number of components $K$ by maximising the BIC score over a set of allowed values $K \leq K_{\max}$. This gives rise to the latent-aware BIC scores

$$\text{BIC}_{\hat{Z}}^{\text{ML}}(\mathcal{H}) = \max_{1 \leq K \leq K_{\max}} \text{BIC}_{\hat{Z}}^{\text{ML}}(\mathcal{H}, K) \quad \text{and} \quad \text{BIC}_{\hat{Z}}^{\text{EM}}(\mathcal{H}) = \max_{1 \leq K \leq K_{\max}} \text{BIC}_{\hat{Z}}^{\text{EM}}(\mathcal{H}, K), \tag{9}$$

where $\text{BIC}_{\hat{Z}}^{\text{ML}}$ uses the oracle estimate and $\text{BIC}_{\hat{Z}}^{\text{EM}}$ the EM one. By appropriate use of the decomposability property, we can show that the criterion of Def. 3.1 holds for the former score, $\text{BIC}_{\hat{Z}}^{\text{ML}}$.

Note that a known result shows that linear additive Gaussian noise models are not identifiable [Shimizu et al., 2006; Pearl, 2009], in the sense that the graphs $X \to Y$ and $X \leftarrow Y$ are then Markov equivalent. In our general case, however, and under mild conditions (see Lemma B.1), our model deviates from this problematic case enough for the identifiability of all models in Fig. 3 to become possible.

**Theorem 3.3** (Local Consistency of $\text{BIC}_{\hat{Z}}^{\text{ML}}$). *Let $\mathcal{D} = \{\mathbf{x}_1, \ldots, \mathbf{x}_r\}$ be observations of random variables $\boldsymbol{X}$, $Y$, such that $\boldsymbol{X}|Y \sim \text{MLR}\left(\mathbf{B}, \boldsymbol{\gamma}, \sigma^2\right)$, with general parameters $\boldsymbol{\theta}$ (see Lemma B.1). Then, out of the structural hypotheses depicted in Fig. 3 the $\text{BIC}_{\hat{Z}}^{ML}$ score of the ground truth hypothesis $\mathcal{G}_{cs}^h$ is asymptotically larger than any of the alternative ones, $\mathcal{G}_{ws}^h$ and $\mathcal{G}_{me}^h$, almost surely.*

As a corollary, GES with the $\text{BIC}_{\hat{Z}}^{\text{ML}}$ can identify between these models.

**Corollary 3.4.** *The latent-aware score $\text{BIC}_{\hat{Z}}^{ML}$ is a consistent scoring criterion.*

In our analysis, however, we have yet to address the discrepancy between the $\text{BIC}_{\hat{Z}}^{\text{ML}}$ that we assumed to be tractable, and the one we actually use above, the $\text{BIC}_{\hat{Z}}^{\text{EM}}$ based on the EM algorithm. Of interest is one particular result of Balakrishnan et al. (2017), which shows that for a Euclidean ball around the optimal parameters of the MLR model, the EM algorithm finds the MLE estimate if it is initialised within this ball. The radius of this ball depends on how well the components of each mixture can be separated, and holds both asymptotically and for the finite sample case.

At the same time, we intuit that when the observations do not fit well with an MLR model, the algorithm will fail to find the correct estimate, resulting in an underestimation, $\text{BIC}_{\hat{Z}}^{\text{EM}} \ll \text{BIC}_{\hat{Z}}^{\text{ML}}$. This assumption would steer the GES algorithm away from selecting the said model, which would instead favor the selection of a correct model. We formalise this intuition as follows.

**Algorithm 1:** DISCOVER A CAUSAL MIXTURE MODEL (CMM)
***
**Input:** Dataset $\boldsymbol{X}$, max. number of mixture components $K_{\max}$
**Output:** Set of latent variables $\boldsymbol{Z}$, causal graph $\mathcal{G}^Z$
***
1 Initialize $\boldsymbol{Z} \leftarrow \emptyset, \mathcal{G}^Z \leftarrow \emptyset, \mathcal{G} \leftarrow \emptyset, T \leftarrow [\,]$;
   // Discover local mixing and graph
2 **while** *not all nodes are ordered* **do**
3    $X_j \leftarrow$ INFERSOURCE$(T, \mathcal{G})$;
4    $\mathcal{G} \leftarrow$ EDGEADDITIONS$(X_j, \mathcal{G})$;
5    $\mathcal{G} \leftarrow$ EDGEPRUNING$(X_j, \mathcal{G})$;
6    **for** *each $k$ in $1, \ldots, K_{\max}$* **do**
7      Using EM, fit $(X_j | \mathbf{Pa}_j = \mathbf{y}) \sim \text{MLR}\left(\mathbf{B}_j, \boldsymbol{\gamma}^j, \sigma^2\right)$ with $k$ components;
8    $Z_j \leftarrow$ mixing assignment with best $\text{BIC}^{\text{EM}}_{\hat{Z}} = \max_{1 \leq K \leq K_{\max}} \text{BIC}^{\text{EM}}_{\hat{Z}}(K)$;
9    $T.\text{APPEND}(X_j)$;
   // Infer global mixing
10 $\boldsymbol{Z}, \mathcal{G}^Z \leftarrow$ INFERMIXING$(\mathcal{G}, \{Z_j\})$;
11 **return** $G, \boldsymbol{Z}$;
***

**Conjecture 3.5.** *The latent-aware score $\text{BIC}^{EM}_{\hat{Z}}$ is a consistent scoring criterion.*

The analysis justifies the use of the latent-aware score $\text{BIC}^{\text{EM}}_{\hat{Z}}$ in score-based algorithms for consistent structure estimation. Inspired by this, we complement our theoretical results with a practical implementation using the $\text{BIC}^{\text{EM}}_{\hat{Z}}$, which we present and evaluate in the remainder of this work.

## 4 Algorithm

Here, we outline an algorithm for discovering *i)* the mixing variables $\boldsymbol{Z}$ and *ii)* the causal graph $\mathcal{G}^Z$. Building on the score consistency that our theory establishes, we propose a joint inference procedure that discovers both components within a score-based framework. At each scoring step, we fit an MLR under a given candidate parent set. Choices for the score-based algorithm include, for example, the GES algorithm [Chickering, 2002] or other score-based causal discovery frameworks. As our main proof-of-concept implementation, we describe how to integrate the latent-aware BIC within the topological order-based framework TOPIC [Xu et al., 2025].

**Causal Mixture Inference and Scoring**    Given a node $Y$ with parents $\text{pa}(Y) \subseteq \boldsymbol{X}$, we score the causal relationship using the latent-aware BIC of Eq. (9). That is, the score accounts for an unknown number of mixtures, where the likelihood takes the form of Eq. (4). To compute the BIC in practice, we need to consider each $1 \leq k \leq K_{\max}$ up to a given hyperparameter $K_{\max}$ and use the Expectation Maximisation (EM) [Bishop, 2006] algorithm to obtain estimates for the MLR parameters; finally, we then use the BIC score of the best such $k$ in our algorithm and the corresponding estimates for the rest of the model parameters. In the special case of source nodes, we infer a GMM similarly using EM. Regarding the number of components $k$, we assume that a reasonably high maximum number of $K_{\max}$ is given that caps the number of mixture components. When MLR model selects $K < K_{\max}$, it is (asymptotically) sure that this estimate of $K$ is the true number of mixture coefficients. When $2 \leq K_{\max} < K^*$, for $K^*$ the true number of clusters, the causal structure would still be correctly inferred, as the capped MLR would still offer a higher likelihood versus any alternative model.

**Causal and Mixing Structure Search**    We discover the causal DAG $\mathcal{G}$ over the observed $\boldsymbol{X}$ by embedding the above inference step into the TOPIC algorithm. Considering a candidate edge $X \to Y$, we also define the score difference $g$ as the difference in BICs before and after adding $X$, $g(X, Y; G) = \text{BIC}_Z(Y \mid \text{pa}(Y) \cup \{X\}) - \text{BIC}_Z(Y \mid \text{pa}(Y))$. TOPIC then constructs the DAG in topological order in the following steps.

     (i) *Source Identification:* Identify $X_j = \arg\max_X \min_Y [g(X, Y; G) - g(Y, X; G)]$ .
     (ii) *Edge Additions:* For each $Y \neq X_t$, add an outgoing edge $X_t \to Y$ if $g(X_t, Y; G) > 0$.
     (iii) *Edge Pruning:* Remove any redundant incoming edges $Z \to X_t$ using $g$.

Following this strategy, we discover a topological order $T$, DAG $\mathcal{G}$ as well as a mixture assignment $Z_i$ for each variable $X_j$ along the topological order. We finally consolidate these. With the reasoning that pairwise estimated $Z_i, Z_l$ exhibit significant overlap when they correspond to the same mixing variable, we measure the similarity of the assignments, here using Adjusted Mutual Information (AMI), and merge variables when the AMI exceeds its expected value [Vinh et al., 2010].

We summarize our approach in Alg. 1.

## 5    Related Work

Discovering causal graphs from observational data is a well-studied problem. Among constraint-based methods is the PC algorithm [Spirtes et al., 2001] using conditional independence (CI) testing, among score-based approaches is GES [Chickering, 2002] using local scoring criteria [Huang et al., 2018]. Both search over the space of Markov Equivalence classes (MECs) over DAGs. LiNGAM [Shimizu et al., 2006] and RESIT [Hoyer et al., 2008] assume non-Gaussianity and independence of regression residuals to orient edges. CAM [Bühlmann et al., 2014] and SCORE [Rolland et al., 2022]/DAS [Montagna et al., 2023] combine topological order estimators with pruning to discover a DAG, while TOPIC [Xu et al., 2025] uses Bayesian or information-theoretic scores for both steps. NOTEARS [Zheng et al., 2018] frames DAG discovery as a continuous optimization problem. Given the potential variance- resp. $R^2$-sortability problems of causal discovery benchmarks [Reisach et al., 2021], Reisach et al., 2023 propose simple baselines exploiting these scoring criteria, VARSORT and $R^2$SORT. Finally, some works relax the common assumption of causal sufficiency and allow for latent confounding variables, such as FCI [Spirtes et al., 1993]. In comparison, with discrete-valued latent variables and special interest in their inference, our setting is more similar to multiple environments.

**Multiple Environments and Interventional Data**  Towards addressing non-i.i.d. data, several works consider different datasets (called environments, contexts, or interventional datasets) that arise from interventions or causal mechanism shifts, but within which all causal mechanisms are fixed [Schölkopf et al., 2021]. Extensions of common causal discovery algorithms to this setting exist [Mooij et al., 2016; Huang et al., 2020; Squires et al., 2020; Mameche et al., 2023]. Common to these methods is that they consider causal discovery in an augmented graphical model with a single latent variable that can be interpreted as the (known) dataset index, respectively, with intervention variables, allowing identifiability of the causal model up to the interventional MEC (iMEC) [Hauser and Bühlmann, 2013; Wang et al., 2017a]. A different line of work, addressing causal discovery in time series [Runge, 2020], considers discovering temporal regimes with changes in causal mechanism, such as RPCMCI [Saggioro et al., 2020]. This approach differs from ours in that it discovers changes in causal mechanisms along the temporal dimension, assuming a temporal ordering of observations.

**Mixture Modeling and Causal Discovery**  Gaussian Mixture Models as well as their conditional counterparts, Mixtures of Linear Gaussian Regressions, are well studied [McLachlan and Peel, 2000; Hennig, 2000]. The idea of reinterpreting clustering and mixture modeling within a causal framework is not new; most works in this area assume a global mixing variable, also termed a latent-class confounder [Mazaheri et al., 2023]. Some works study for example the bivariate cases [Hu et al., 2018], discrete variables [Mazaheri et al., 2023] and causal inference settings [Kim et al., 2024; Mazaheri et al., 2025]. Most closely related to ours is the work of Kumar et al., 2024 which proposes Mixture-UT-IGSP (MIXUTIGSP), combining a GMM with adequate component selection with UT-IGSP. Kumar et al., 2024 give insight into the sample complexity of identifying interventional mixtures, as well as establish identifiability of the iMEC in this setting using the two-stage approach. We will evaluate against this approach in the next section.

**Causal Identifiability**  Methods exist to identify linear additive Gaussian noise models with equal variance, under further conditions, such as non-perfect dependencies, and what is more, also when other constraints are added (e.g., that the coefficients are all less than the unit) [Peters and Bühlmann, 2013]. Our identification results do not rely on these specific assumptions; instead, we base identification on the effects of the latent mixing variables and the asymmetries they introduce in the causal directions. The results are thus independent of the approach of restricting the functional model with further assumptions, and can address scenarios where these assumptions do not hold.

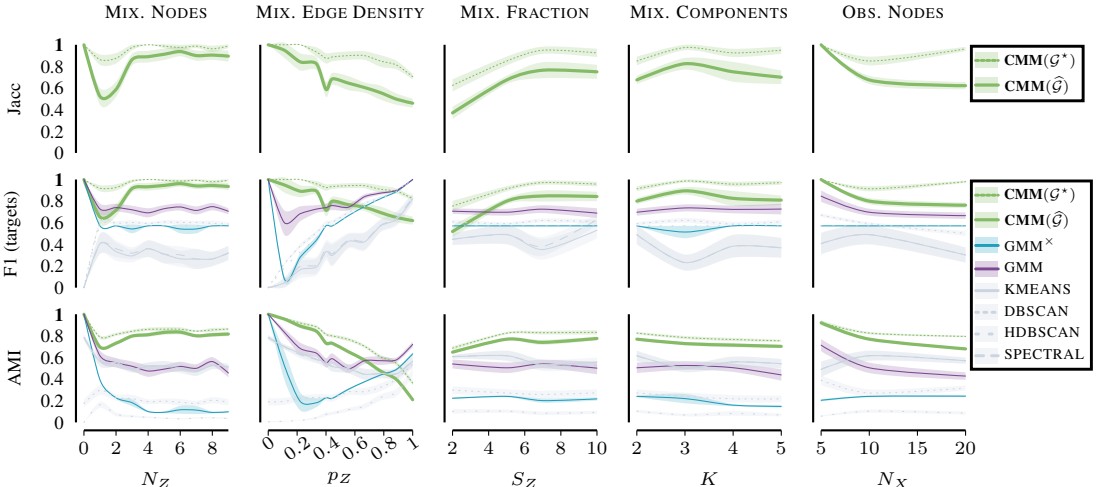

Figure 4: *Discovering Mixing Structure.* In synthetically generated CMMs, we evaluate the quality of the recovered mixing structure, evaluating the affected observed variables (F1 (target)), variable sets affected by the same latent variable (Jacc) as well as per-node mixing assignments (AMI).

## 6 Evaluation

We evaluate our approach on two main questions,

(i) *Discovering Mixing Structure*: Given a causal graph, can our approach accurately discover the underlying mixing structure, including the number of mixing variables, the number of their components, assignments, and sets of targeted observed variables?

(ii) *Discovering Causal Structure*: When the causal graph is unknown, can our approach recover the mixing structure as well as the causal graph?

We address the above questions on synthetic and real-world data.

**Experimental Setup** We generate data according to our assumptions by drawing Erdős Rènyi DAGs using $X_j | \mathbf{Pa}_j \sim \mathrm{MLR}\left(\mathbf{B}_j, \boldsymbol{\gamma}^j, \sigma^2\right)$ with Gaussian additive noise $N_j \sim \mathcal{N}(0, 1)$ and ensuring that the linear coefficients $\mathbf{B}_j$ are bounded away from zero, $\boldsymbol{\beta}_{jk} \in [-1, -0.25] \cup [0.25, 1]$ and similarly from one another. To avoid issues related to Var-sortability and $R^2$-sortability [Reisach et al., 2021] we generate an internally standardized structural causal model (iSCM) [Ormaniec et al., 2024].

The experiments address the effect of several parameters: the number of observed $N_X \in \{5, \ldots 20\}$ and latent variables $N_Z \in \{0, \ldots, 10\}$, number of latent classes $K \in \{2, \ldots, 5\}$, fraction of observed variables $p_Z \in [0, 1]$ affected by mixing, dag density $p_{\mathcal{G}} \in [0, 1]$, sample size $S \in \{200, \ldots 1000\}$ and a parameter controlling the size of the samples in each group $S_Z \in \{2, \ldots, 10\}$; for example, for $K=2$, we uniformly at random draw $\frac{S}{S_Z}$ samples belong class 0, otherwise assign class 1. By default, we show results for $N_X = 10, N_Z = 4, K = 2, p_Z = 0.5, p_{\mathcal{G}} = 0.4, S = 500, S_Z = 5$. For a detailed description of the data generation setup, see Appendix D.

### 6.1 Discovering Mixing Structure

We assess the quality of mixing assignments $Z_i$ using the Adjusted Mutual Information (AMI) averaged over the estimated vs. true assignments for each observed node $X_j$ in a simulated graph $\mathcal{G}$, and the mixing structure in $\mathcal{G}^Z$ using F1 scores (binary edge existence $Z_i \rightarrow X_j$) and Jaccard indices (set overlap of each $Z_i$'s observed targets). We include simple baselines that apply mixture modeling without causal information for comparison.

In Fig. 4, we observe reliable recovery of mixing assignments (AMI) and their targets (F1, Jacc) for our approach given the true graph (dashed green) and for the full framework (solid green). Applying a clustering baseline such as GMM over all nodes in the graph, as in [Kumar et al., 2024] (blue, $\times$), is misspecified in case $N_Z > 1$, as per-node assignments are distinct, hence not surprisingly,

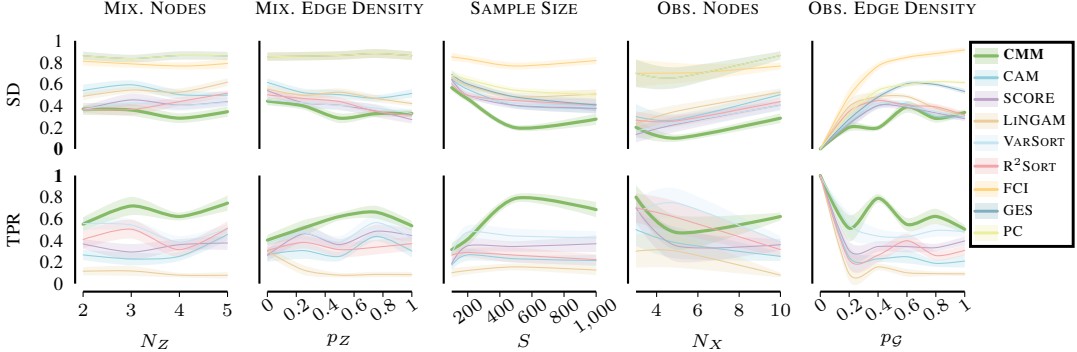

Figure 5: *Discovering Causal Graphs.* In synthetically generated CMMs, we evaluate the quality of the causal DAG over the observed variables in terms of the separation distance (SD) and true positive rate over directed edges (TPR).

performance worsens as $N_Z$ increases (left). Given this, we also apply the models to each node in turn (purple, gray), where gray variants show different instantiation choices. These perform not much better than random splitting in most cases (AMI, F1) and with accurate set recovery (Jacc).

## 6.2 Discovering Causal Structure

Next, we evaluate the learned causal DAG resp. CPDAG $\mathcal{G}$ against the ground truth $\mathcal{G}^\star$. We consider the separation-based distance metrics proposed by Wahl and Runge, 2024, measuring among others the Separation Distance (SD) over graphs. To demonstrate the strength of our approach in discovering edge orientations in particular, we also compute orientation F1 scores and report true positive rates (TPR). Additional metrics of interest are postponed to the Appendix. Baselines include a range of causal discovery methods, CAM, SCORE, LiNGAM, FCI, GES, PC, VARSORT, and $R^2$SORT.

While some baselines perform reasonably well in mixed settings, CMM maintains high accuracy across both structural and causal direction evaluations (Figure 5). We note that Mixture-UTIGSP is not meaningfully applicable here, as there may be multiple mixing variables and no well-defined "observational" part of the dataset as required as input to UTIGSP; thus we will instead consider an experimental setting with a single mixing variable.

**Case Study: Mixtures of Interventions** Next, we replicate the experimental setup of Kumar et al., 2024 using their data generators to evaluate performance in the mixtures-of-interventions setting. As Figure 6 shows, the GMM used in Mixture-UTIGSP performs well in recovering mixture assignments as expected, given that its modeling assumptions are met in this setting. Our method performs comparably in mixture assignment recovery and recovers the causal graph more reliably (Figure 6 bottom). We also noticed a competitive performance of VARSORT in this dataset suggesting potential Var-sortabity which may inflate the performance of the discovery approaches.

**Case Study: Flow Cytometry Data** Finally, we investigate the real-world flow cytometry dataset curated by Sachs et al., 2005. The dataset originates from single-cell protein-signalling samples of the human immune response system, where each of the 11 observed variables corresponds to the activity of one compound of interest: either a protein or a phospholipid. The dataset contains (among others) 5 experimental conditions, in each of which a particular molecular modifier has been applied

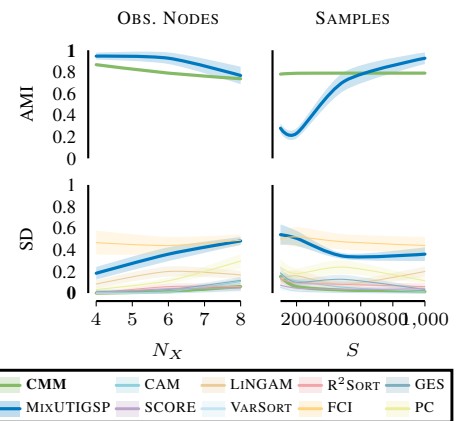

Figure 6: *Mixtures of Interventions.* For data generated from an interventional mixture, we show the quality of mixture separation (AMI, cf. Fig. 4) and causal DAG discovery (SD, cf. Fig. 5).

| | MIXED/IV. SAMPLES (AMI) | | | | | CAUSAL GRAPH | | |
|---|---|---|---|---|---|---|---|---|
| TARGET | CMM* | **CMM** | MIXUTIGSP | | METRIC | **CMM** | MIXUTIGSP |
| Akt | 0.02 | 0.04 | 0.04 | | TP | 5 | 0 |
| PKC | 0.26 | 0.43 | **0.49** | | FP | 10 | 2 |
| PIP2 | 0.10 | **0.10** | 0.03 | | FN | 5 | 10 |
| Mek | 0.20 | **0.20** | 0.14 | | SD | **0.25** | 0.75 |
| PIP3 | 0.00 | 0.00 | 0.00 | | S/C | **0.27** | 0.57 |

Table 1: *Mixing discovery* as in Fig. 4, for the data by Sachs et al., 2005.

Table 2: *Causal discovery* as in Fig. 5, for the data by Sachs et al., 2005.

to the cells, such that the activity of exactly one of the 11 compounds of interest is affected; this results in a known change of the measured activity for the corresponding compound. As in previous analyses [Wang et al., 2017b], we combine the data from all experimental conditions into a larger dataset of size 5846, and do not disclose their origin to the algorithm. Hence, in the pooled data, each variable is affected by a latent one.

For example, considering the node "'Akt'", there are two mixture components: one is the experimental condition where the so-called Akt inhibitor was applied, directly inhibiting Akt activation; the other mixture component comprises the remaining samples where Akt is in its baseline condition. In particular, the variables Akt, PKC, PIP2, Mek, and PIP3 were manipulated. We test how well, for each given target node, its intervened subsample can be recovered (AMI). The dataset split found by Mixture-UTIGSP appears to match PKC best, similar for our per-variable splits which match PIP2 and MEC slightly better (Table 1). Still, neither method reaches convincing AMI, consistent with the observations [Kumar et al., 2024; Squires et al., 2020] that the interventional targets are difficult to identify in this dataset. In terms of causal discovery, our approach performs better in terms of separation distances (SD, S/C) albeit discovering a number of spurious (FP) edges (Table 2). All other baselines report similarly high false positive directions (10+ FP) with the exception of PC (2 FP). Additionally, some edge directions in the ground truth such as Raf $\rightarrow$ Mek are subject to debate. We discover the Mek $\rightarrow$ Raf direction which has been previously discussed as agreeing better with the given data [Mooij et al., 2016].

In summary, the experiments give empirical support for effectively recovering the mixing structure and the causal structure. In Appendix E, we additionally (i) compare the GES to the TOPIC instantiation; (ii) study the effect of latent mixing variables on causal discovery algorithms; and (iii) compare functional forms both in the data generation as well as during scoring.

## 7  Conclusion

We address the problem that real-world datasets rarely come from a fixed data-generating process, but could be a combination of subpopulations with distinct causal mechanisms. This gives rise to a causal mixture model in the sense that each effect given its causes results from a finite mixture of linear regressions. We characterize this model in our work and establish theoretical results showing that consistent scoring criteria, such as the latent-aware BIC, allow causal identifiability. Complementary to these insights, we propose a practical implementation for causal discovery using this score which we demonstrate to have strong empirical performance in various settings.

In the real-world case study [Sachs et al., 2005], while the approaches recover the causal graph structure to an extent, we observed limited ability of both the CMM and Mixture-UTIGSP to discover the intervention targets, likely due to the strong assumption of linearity. As our proposed approach can in principle easily incorporate richer models, future work could study scoring criteria and theoretical guarantees for non-linear mixtures, encouraged by a proof-of-concept experiment that we provide in Appendix E. Future work could also study how to further interpret the meaning of a given $Z$, especially when it points to previously unknown states, for example, by exploring explainability approaches such as LIME [Ribeiro et al., 2016] or Shapley values [Lundberg and Lee, 2017]. Furthermore, invoking Expectation Maximization at every scoring step comes with drawbacks regarding the scalability of our method, especially when relying on random restarts, therefore subsequent analysis could devise algorithmic approaches to reduce this overhead.

## Acknowledgements

We would like to thank the anonymous reviewers for their helpful feedback.

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

# A  Preliminaries

In this section, we provide a brief overview of helpful preliminary concepts that, although relevant to our analysis, we assume to be widely familiar in the main target audience of our work; for brevity and clarity, we therefore chose to postpone them away from the main manuscript and into this appendix.

## A.1  Structural Causal Models

Given random variables $\boldsymbol{X} = \{X_1, \ldots, X_n\}$ we can encode the underlying data generating process (DGP) as a structural equation model (SEM) [Koller and Friedman, 2009]; this model encodes a set of hypotheses on this process in the form of one functional dependency $f_j$ for each random variable $X_j \in \boldsymbol{X}$, so that

$$X_j = f_j(\boldsymbol{X}_j) \qquad \text{with } \boldsymbol{X}_j \subseteq \boldsymbol{X} \setminus \{X_j\}. \tag{10}$$

Of special interest is a structural causal model (SCM) [Bollen, 1989], which is a particular kind of an SEM with additional assumptions that allow it to also model the causal mechanisms of the DGP. Here, the set of random variables $\boldsymbol{X}$ is extended to also include random unobserved variables $\mathbf{U} = \{U_1, \ldots, U_n\}$, which play the role of noise. Hence, each functional dependency takes the form

$$X_j = f_j(\boldsymbol{X}_j, U_j) \qquad \text{with } \boldsymbol{X}_j \subseteq \boldsymbol{X} \setminus \{X_j\} \text{ and } U_j \in \mathbf{U}. \tag{11}$$

To further study SCMs, we need to establish their correspondence with causal graphs [Pearl, 2009].

## A.2  Causal Graphs

Consider a set of random variables $\boldsymbol{X} = \{X_1, \ldots, X_n\}$ that follow a distribution $\mathcal{L}_{\boldsymbol{X}}$ that has a joint probability density $p_{\boldsymbol{X}}$ with respect to some appropriate measure, and an (arbitrary) total ordering $X_1 < X_2 < \cdots < X_n$. Then the joint probability density factorises as

$$p_{\boldsymbol{X}}(\mathbf{x}) = \prod_{X_j \in \boldsymbol{X}} p_{X_j | \mathbf{Y}_j}(x, \mathbf{y}_j), \qquad \text{where } \mathbf{Y}_j \subseteq \{X_1, \ldots, X_j\} \tag{12}$$

and $\mathbf{y}_j$ are those values out of $\mathbf{x}$ corresponding to the same indices as $\mathbf{Y}_j$. This comes as a direct result of the chain rule and the conditional independence rules. Any such factorisation can be represented as a (fully) directed acyclic graph (DAG) $\mathcal{G} = (\boldsymbol{X}, E)$ with nodes the random variables $\boldsymbol{X}$ and edges the set $E = \cup_{j=1}^{n} \{i \to j | X_i \in \mathbf{Y}_j\}$. In other words, in this graph we add an edge to the dependent variable $X_j$ from each variable in the corresponding conditioning set $\mathbf{Y}_j$ that appears in each factor $p_{X_j | \mathbf{Y}_j}$ of Eq. (12). We further make this relation explicit, by instead writing $\mathbf{Pa}_j = \mathbf{Y}_j$ to indicate that the conditioning set $\mathbf{Y}_j$ serves as the set of direct parents of node $X_j$ in $\mathcal{G}$. Such a graph is called a *Bayesian network* [Koller and Friedman, 2009] and allows for a visual representation of all those independencies that are implied solely from this factorisation of the joint density and irrespective of the form of each factor.

These independencies can be read from the graph in terms of the d-separation [Pearl, 2009].

**Definition A.1** (d-separation). For any pairwise disjoint subsets $\mathbf{U}, \mathbf{V}, \mathbf{W} \subseteq \boldsymbol{X}$, and $\mathbf{U}, \mathbf{V} \neq \emptyset$, it is

$$\mathbf{U} \perp\!\!\!\perp \mathbf{V} | \mathbf{W} \iff \text{all paths from any variable in } U \text{ to any of } V \text{ are blocked by } W. \tag{13}$$

We call a path blocked if it either

- traverses a section $\to V \to$, $\leftarrow V \leftarrow$ or $\leftarrow V \to$ for some variable $V \in W$, or
- traverses a section $\to V \leftarrow$ where neither $V$ nor any of its descendants are contained in $W$.

Hence, a Bayesian network $\mathcal{G}$ can be seen as a description of an entire family of distributions that fulfill a given set of conditional independencies. When a distribution $\mathcal{L}_{\boldsymbol{X}}$ exhibits all the conditional independencies that one can read from the graph $\mathcal{G}$, we call $\mathcal{G}$ an I-map of $\mathcal{L}_{\boldsymbol{X}}$ and write $\mathcal{L}_{\boldsymbol{X}} \in I_{\mathcal{G}}$.

In other words, the I-map defines an equivalence relation among all DAGs via the relation $\mathcal{G} \equiv_M \mathcal{G}' \iff I_{\mathcal{G}} = I_{\mathcal{G}'}$, of which each equivalence class is called the *Markov equivalence class* (MEC). When, in addition, each of the factors in the factorisation of Eq. (12) correspond to a functional dependency of an SCM, we call the resulting DAG causal.

A graphical representation of MECs can be given through the common notion of *completed partially directed acyclic graphs* (CPDAGs) [Chickering, 2002] also known under other names such as maximally oriented graphs [Meek, 1997]. A partially directed graph (PDAG) $\mathcal{P}$ contains both undirected and directed edeges, and can be associated to an equivalence class $\mathcal{M}(\mathcal{P})$ with $\mathcal{G} \in \mathcal{M}(\mathcal{P})$ if and only if $\mathcal{G}, \mathcal{P}$ have the same skeleton and v-structures. The notion of completion of PDAGs allows for representing equivalence classes uniquely. To this end, for a given equivalence class $\mathcal{M}$ one distinguishes between *compelled* edges with the same directionality in every member of $\mathcal{M}$, and *reversible* edges otherwise. The completed PDAG $\mathcal{P}$ for $\mathcal{M}$ is then the one having a directed edge for every compelled edge in $\mathcal{M}$, and an undirected edge for every reversible edge in $\mathcal{M}$.

### A.3 SCMs and causal graphs

We now return to the assumptions implicit in an SCM.

**Assumption 2** (Causal Interpretation). Each functional dependency $f_j$ corresponds to a true causal mechanism in the data, with $\boldsymbol{X}_j$ being the direct causes of the direct effect $X_j$.

As a corollary, the corresponding causal graph can have no recurrence.

**Assumption 3** (Orientation and Acyclicity). The causal graph of an SCM is a DAG.

Hence, we can once again identify the direct causes of each effect $X_j$ with its parents in the corresponding DAG, $\boldsymbol{X}_j = \mathbf{Pa}_j$.

**Assumption 4** (Exogeneity of Noise). The random variables $\mathbf{U}$ are exogenous; that is, there are no edges $X_j \to U_i$, for any $X_j$ and any $U_i$.

**Assumption 5** (Independence of Noise). The random variables $\mathbf{U}$ are mutually independent.

In our work, in particular, we consider that part of the noise for each effect $X_j$ is the latent variable $\mathrm{La}_j$, in addition to the typical $U_j$. Hence, even though for the set $\mathbf{U}$ we do make Assumption 5, we note that this fails to hold for the entire set of exogenous noise sources, which, in our case, is further extended to include all $\boldsymbol{Z}$. As a result, to be more rigorous, we make claims to find the CPDAG corresponding to the conditional distribution $\mathcal{L}_{\boldsymbol{X}|\boldsymbol{Z}}$, rather than the marginal $\mathcal{L}_{\boldsymbol{X}}$.

We remark that, if one is interested in the entire marginal $\mathcal{L}_{\boldsymbol{X}}$, this can be achieved as a two-level algorithm, in which we first use our presented method to infer the Markov equivalence class of $\mathcal{L}_{\boldsymbol{X}|\boldsymbol{Z}}$, and then use the inferred values of each $\mathrm{La}_j$ to identify which variables correspond to the same underlying latent $Z_i$, and finally performing causal structure inference over the values of $\boldsymbol{Z}$ to complete the Markov equivalence class of $\mathcal{L}_{\boldsymbol{X}}$.

In the scope of our work, we hence focus on the conditional $\mathcal{L}_{\boldsymbol{X}|\boldsymbol{Z}}$. In the following section we formally show that we can recover the corresponding underlying Markov equivalence class, as represented by a CPDAG $\mathcal{G}$. Specifically, we next move to formally showing our main claims in Corollary 3.4 on using the proposed latent-aware BIC as a consistent scoring criterion for this purpose.

## B  Proof of Theorem 3.3

In this section we elaborate on the theoretical justification of our method. For this, we make use of the relaxed space constraints to break down our analysis in finer steps. We first show that under mild conditions, the MLR distribution does not degenerate to a Gaussian.

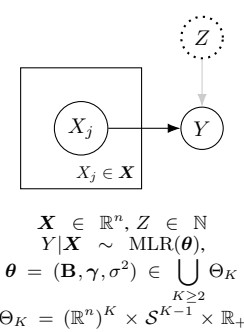

We hence treat the parameters $\boldsymbol{\theta}$ as random variables and assume each MLR distribution in the data generating process to be drawn as follows. First, a finite number $K \geq 2$ of components is arbitrarily fixed; then the tuple $\boldsymbol{\theta} = (\mathbf{B}, \boldsymbol{\gamma}, \sigma^2) \in \Theta_K$ is drawn, where $\Theta_K = (\mathbb{R}^n)^K \times \mathcal{S}^{K-1} \times \mathbb{R}_+$ is the parameter space of the model with $K$ components and $\mathcal{S}^K$ the $K$-dimensional probability simplex.

Out of this tuple, the linear coefficients are assumed to be drawn from a prior $\mathbf{B} \sim \mu_{\mathbf{B}}^K$, the mixture coefficients from $\boldsymbol{\gamma} \sim \mu_{\boldsymbol{\gamma}}^K$, and a positive variance $\sigma^2 > 0$ from an arbitrary distribution.

Also let $\lambda_{\mathbf{B}}^K$ be the push-forward of the Lebesgue measure on $\mathbb{R}^{n \times K}$

$$
\begin{aligned}
\boldsymbol{X} &\in \mathbb{R}^n, Z \in \mathbb{N} \\
Y|\boldsymbol{X} &\sim \mathrm{MLR}(\boldsymbol{\theta}), \\
\boldsymbol{\theta} = (\mathbf{B}, \boldsymbol{\gamma}, \sigma^2) &\in \bigcup_{K \geq 2} \Theta_K \\
\Theta_K &= (\mathbb{R}^n)^K \times \mathcal{S}^{K-1} \times \mathbb{R}_+
\end{aligned}
$$

Figure B.1: The parameters of the MLR distribution.

through the homeomorphism of that space with $(\mathbb{R}^n)^K$ and $\lambda_{\boldsymbol{\gamma}}^K$ the Lebesgue measure on $\mathcal{S}^{K-1}$.

**Lemma B.1** (Non Gaussianity of Direct Effect). *Let $\boldsymbol{X} \in \mathbb{R}^d$, $Y \in \mathbb{R}$ be random variables such that $Y|\boldsymbol{X} \sim \text{MLR}\left(\mathbf{B}, \boldsymbol{\gamma}, \sigma^2\right)$, with parameters $\boldsymbol{\theta} = (\mathbf{B}, \boldsymbol{\gamma}, \sigma^2) \in \Theta = \cup_{K \geq 2}\Theta_K$. Let the parameter prior $\mu_\Theta$ be $\sigma$-additive; let for each $\Theta_K$ the marginal priors of the linear coefficients and the mixture coefficients[1] be absolutely continuous with respect to the corresponding Lebesgue measure, that is, $\mu_{\mathbf{B}}^K \ll \lambda_{\mathbf{B}}^K$ and $\mu_{\boldsymbol{\gamma}}^K \ll \lambda_{\boldsymbol{\gamma}}^K$.*
*Then the distribution of $Y|\boldsymbol{X}$ is almost surely not a Gaussian.*

*Proof.* Fix a $K \geq 2$. We first show that the MLR distribution with density

$$p_{Y|\boldsymbol{X}}^{\text{MLR}}(y, \mathbf{x}; \mathbf{B}, \boldsymbol{\gamma}, \sigma^2) = \sum_{k=1}^{K} \gamma_k p_X^{\mathcal{N}}(x; \boldsymbol{\beta}_k^\top \mathbf{y}, \sigma^2) \tag{14}$$

almost never degenerates into a Gaussian.

The alternative can only happen under either of the following two conditions:

1. there is only one (active) component in the mixture, or
2. each active component has the same linear coefficients.

For the former condition to hold, it must be that for the drawn $\boldsymbol{\gamma}$ it is $\gamma_k = 1$ for some $1 \leq k \leq K$. The probability of drawing such a $\boldsymbol{\gamma}$ is equal to $\mu_{\boldsymbol{\gamma}}^K(E_K)$ where

$$E_K = \{\boldsymbol{\gamma} \in \mathcal{S}^{K-1} \mid \exists k, \, 1 \leq i \leq K, \, \gamma_k = 1\} \tag{15}$$

is the set of the $K$ extremal points of $\mathcal{S}^{K-1}$. However, $E_K$ is a finite subset of $\mathcal{S}^{K-1}$, and so $\lambda_{\boldsymbol{\gamma}}^K(E_K) = 0$; this, implies that $\mu_{\boldsymbol{\gamma}}^K(E_K) = 0$, since $\mu_{\boldsymbol{\gamma}}^K \ll \lambda_{\boldsymbol{\gamma}}^K$.

To study the latter condition, we treat the domain of the linear coefficients as $\mathbf{B} \in \mathbb{R}^{n \times K}$, which is homeomorphic to $(\mathbb{R}^n)^K$. Denote $V_K$ the set of matrices with all columns equal. This set forms a linear subspace $V_K \subseteq \mathbb{R}^{n \times K}$, as it contains the zero matrix and is closed under scalar multiplication and addition. Additionally, this linear subspace $V_K$ is a proper subspace of $\mathbb{R}^{n \times K}$, as the latter contains matrices with unequal columns, which are not part of $V_K$. Therefore, the Lebesgue measure of this subspace vanishes, $\lambda_{\mathbf{B}}^K(V_K) = 0$, which in turn implies that $\mu_{\mathbf{B}}^K(V_K) = 0$, due to $\mu_{\mathbf{B}}^K \ll \lambda_{\mathbf{B}}^K$.

We now consider the pre-images $\bar{V}_K = \pi_{\mathbf{B}|K}^{-1}(V_K)$ and $\bar{E}_K = \pi_{\boldsymbol{\gamma}|K}^{-1}(E_K)$ of these sets, where $\pi_{\mathbf{B}|K} : \Theta_K \to \mathbb{R}^{n \times K}$ and $\pi_{\boldsymbol{\gamma}|K} : \Theta_K \to \mathcal{S}^{K-1}$ are the Cartesian projections from the space of parameter tuples in $\Theta_K$ to that of the space of linear coefficients and mixing coefficients, respectively. Further, since $\Theta = \bigcup_{K \geq 2} \Theta_K$, it is also that $\bar{V}_K, \bar{E}_K \in \Theta$. However, since the measure of both $V_K$ and $E_K$ vanished in the corresponding marginal measures, $\mu_{\mathbf{B}}^K, \mu_{\boldsymbol{\gamma}}^K$, respectively, it must also be that $\mu_\Theta(\bar{V}_K) = \mu_\Theta(\bar{E}_K) = 0$.

Hence, the probability that for a given $K$ the distribution degenerates is equal to $\mu_\Theta(\bar{D}_K)$, which we define as $\bar{D}_K := \bar{V}_K \cup \bar{E}_K$. Now, by the union bound $\mu_\Theta(\bar{D}_K) \leq \mu_\Theta(\bar{V}_K) + \mu_\Theta(\bar{E}_K) = 0$.

Finally, we can bound the probability that the MLR distribution degenerates into a Gaussian for an arbitrarily drawn number of components. Let $\bar{D} \subseteq \Theta$ be the parameters of the MLR distribution that correspond to the cases that degenerate into a Gaussian, for which it is $\bar{D} = \bigcup_{K \geq 2} \bar{D}_K$. By the $\sigma$-additivity of $\mu_\Theta$ we get

$$\mathbb{P}(\{\text{MLR } degenerates \text{ into } Gaussian\}) = \mu_\Theta(\bar{D}) = \mu_\Theta\left(\bigcup_{K \geq 2} \bar{D}_K\right) = \sum_{K \geq 2} \mu_\Theta(\bar{D}_K) = 0, \tag{16}$$

which concludes the proof. $\qquad\square$

To show identifiability, we need to be able to distinguish, under the true hypothesis $Y|\boldsymbol{X} \sim \text{MLR}$, between all competing hypotheses of Fig. B.2.

**Theorem 3.3** (Local Consistency of $\text{BIC}_{\hat{Z}}^{\text{ML}}$). *Let $\mathcal{D} = \{\mathbf{x}_1, \ldots, \mathbf{x}_r\}$ be observations of random variables $\boldsymbol{X}$, $Y$, such that $\boldsymbol{X}|Y \sim \text{MLR}\left(\mathbf{B}, \boldsymbol{\gamma}, \sigma^2\right)$, with general parameters $\boldsymbol{\theta}$ (see Lemma B.1). Then, out of the structural hypotheses depicted in Fig. 3 the $\text{BIC}_{\hat{Z}}^{\text{ML}}$ score of the ground truth hypothesis $\mathcal{G}_{cs}^h$ is asymptotically larger than any of the alternative ones, $\mathcal{G}_{ws}^h$ and $\mathcal{G}_{me}^h$, almost surely.*

---

[1]Note that a weaker condition is needed for the mixture coefficients, namely that at least two components have positive probability. This more general requirement, however, is already general enough and easier to treat.

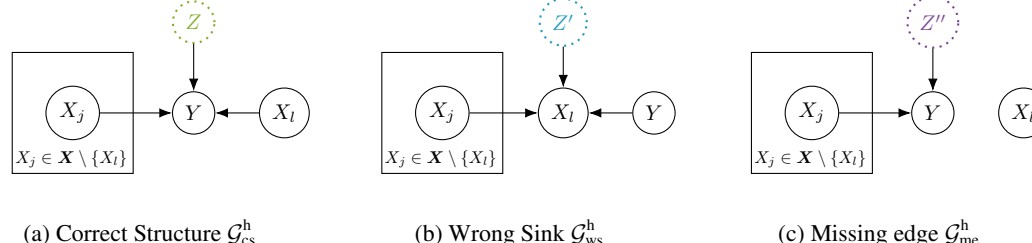

(a) Correct Structure $\mathcal{G}_{\mathrm{cs}}^{\mathrm{h}}$  (b) Wrong Sink $\mathcal{G}_{\mathrm{ws}}^{\mathrm{h}}$  (c) Missing edge $\mathcal{G}_{\mathrm{me}}^{\mathrm{h}}$

Figure B.2: *Identifiable Cases.* Under mild assumptions, when $Z$ has at least two mixing components, all of the shown DAGs are identifiable.

*Proof.* This claim builds on the established properties of the vanilla BIC score. Here, we treat $\boldsymbol{X} \setminus \{X_l\}$ as nuisance parameters, and we focus on single edge modifications, between the sink of each structural hypothesis and the node on the right of each depiction.

We first consider the pair $\mathcal{G}_{\mathrm{cs}}^{\mathrm{h}}$ and $\mathcal{G}_{\mathrm{ws}}^{\mathrm{h}}$. In this case, since the $\mathrm{BIC}_{\hat{Z}}^{\mathrm{ML}}$ is based on the Maximum Likelihood Estimate (MLE) estimates $\hat{\boldsymbol{\theta}}$ of the true parameter values $\boldsymbol{\theta}$, and the MLE estimate is asymptotically unbiased, then the the correct model $\mathcal{G}_{\mathrm{cs}}^{\mathrm{h}}$ and the one arising from the alternate hypothesis $\mathcal{G}_{\mathrm{ws}}^{\mathrm{h}}$ have the same number of parameters, while at the same time the likelihood under $\mathcal{G}_{\mathrm{cs}}^{\mathrm{h}}$ is larger than that of $\mathcal{G}_{\mathrm{ws}}^{\mathrm{h}}$. Hence, in this case the $\mathrm{BIC}_{\hat{Z}}^{\mathrm{ML}}$ value is an increasing function of only the likelihood, and hence it must be that also $\mathrm{BIC}_{\hat{Z}}^{\mathrm{ML}}(\mathcal{G}_{\mathrm{cs}}^{\mathrm{h}}) > \mathrm{BIC}_{\hat{Z}}^{\mathrm{ML}}(\mathcal{G}_{\mathrm{ws}}^{\mathrm{h}})$.

For the pair $\mathcal{G}_{\mathrm{cs}}^{\mathrm{h}}$ and $\mathcal{G}_{\mathrm{me}}^{\mathrm{h}}$ the number of parameters in the latter hypothesis is smaller than that of the $\mathcal{G}_{\mathrm{cs}}^{\mathrm{h}}$. Under similar reasoning as in Lemma B.1, we can claim that $Y \not\perp\!\!\!\perp X_l$ almost surely. The rest follows from established asymptotic behaviour of $\mathrm{BIC}_{\hat{Z}}^{\mathrm{ML}}$ as a special case of BIC, due to the decomposability property that $\mathrm{BIC}_{\hat{Z}}^{\mathrm{ML}}$ inherits from BIC. $\qquad\square$

Using this result, we can extend the local consistency of $\mathrm{BIC}_{\hat{Z}}^{\mathrm{ML}}$ to its global consistency.

**Corollary 3.4.** *The latent-aware score* $\mathrm{BIC}_{\hat{Z}}^{ML}$ *is a consistent scoring criterion.*

*Proof.* By considering any sequence of appropriate single edge modifications between adjacent structural hypotheses as in Fig. B.2, we can extend the global consistency of BIC to that of $\mathrm{BIC}_{\hat{Z}}^{\mathrm{ML}}$ Chickering, 2002. $\qquad\square$

## C  Implementation Considerations

We note that our main theory covers the Greedy Equivalent Search (GES) algorithm. In our implementation, however, we have used TOPIC [Xu et al., 2025], a more recent greedy score-based search that has similar guarantees to GES, and when similar requirements are met by the used score. Hence, we replace within TOPIC our proposed $\mathrm{BIC}_{\hat{Z}}^{\mathrm{EM}}$ score, to thus derive $\mathrm{TOPIC}_{\mathrm{BIC}}$, and here analyse the two algorithms.

We first assume access to the MLE oracle. Then, although the output of both algorithms lies on the domain of all CPDAGs over $\boldsymbol{X}$, $\mathrm{TOPIC}_{\mathrm{BIC}}$ builds on TOPIC, which is both asymptotically and practically more efficient than GES. Indeed, in each iteration, it first limits the candidate hypotheses from the set of all representatives of the Markov equivalence classes which perform a single-edge modification from the current best, to only those which differ with respect to the most likely modified source. Formally, the combination of these two steps are two subsequent maximisations over exactly the same domain (of all hypotheses with single edge modifications), only performed first over the possible sources, and then over the rest of the hypotheses, conditioned on the chosen source. Hence, at each iteration, the asymptotic greedy optimum is the same in both algorithms.

To see the asymptotical consistency, in the particular assumptions of our causal model, we treat two different cases. First, when no latent variable affects the result, $\mathrm{TOPIC}_{\mathrm{BIC}}$ would have similar ease to detect edge additions as in the case of TOPIC/GES. In the case that a the true structure is an Mixture

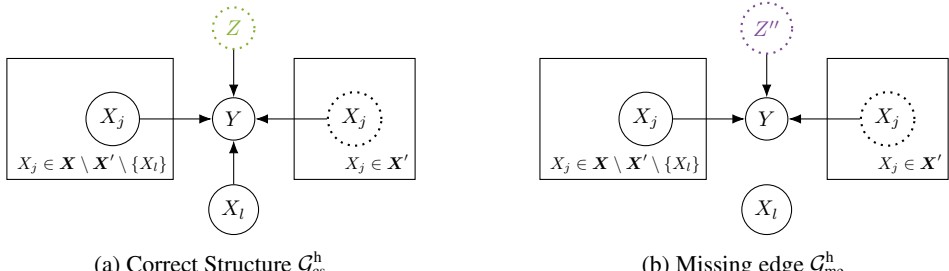

(a) Correct Structure $\mathcal{G}^{\mathrm{h}}_{\mathrm{cs}}$           (b) Missing edge $\mathcal{G}^{\mathrm{h}}_{\mathrm{me}}$

Figure C.1: Decomposability of the MLR model in the intermediate stages of TOPIC$_{\mathrm{BIC}}$. The so-far-undiscovered edges in $\boldsymbol{X}' \subseteq \boldsymbol{X}$ are akin to noise that affects both cases equally.

of Linear Regressions (MLR) model, we can consider the sequence of edge modifications that the TOPIC$_{\mathrm{BIC}}$ algorithm would produce. Within this sequence, we can revisit the cases of Fig. B.2, and notice that when a subset $\boldsymbol{X}' \subset \boldsymbol{X}$ is not yet discovered in the structure of the algorithm, the effects of the so-far-undicovered variables $\boldsymbol{X}'$ can be seen as added noise, which equally affects an appropriate hypothesis $\mathcal{G}^{\mathrm{h}}_{\mathrm{cs}}$ and $\mathcal{G}^{\mathrm{h}}_{\mathrm{me}}$, as shown in Fig. C1.

Hence, intuitively, we expect a point at which one of the edges of the type $X_l \to Y$ would be added to the model, until all parents $\boldsymbol{X}$ will be discovered. Finally, we posit that the practical use of $\mathrm{BIC}^{\mathrm{EM}}_{\hat{Z}}$ in lieu of $\mathrm{BIC}^{\mathrm{ML}}_{\hat{Z}}$ is equally affecting both frameworks, as long as Conjecture 1 holds.

# D    Detailed Evaluation

**Experimental Setup**    We give a more detailed description of our synthetic data generation here. In iteration $i \in \{1, \ldots N_I\}$ of each experiment, we randomly sample a DAG $\mathcal{G}$ over $N_X := |\boldsymbol{X}|$ observed variables under an Erdős Rènyi model with edge density $p_{\mathcal{G}} \in [0, 1]$. In addition, we draw $N_Z := |\boldsymbol{Z}|$ latent mixing variables $Z_i \sim \mathrm{Categorical}(\boldsymbol{\gamma}^i)$ with $Z_i \in \{1, \ldots, K_i\}$, where we fix all $K_i =: K$ to the same hyperparameter $K$. We then sample a set of so-called mixing targets $\boldsymbol{T} = \{X_j \mid \exists Z_i : \mathrm{La}_j = Z_i\} \subseteq \boldsymbol{X}$ where $X_j \in \boldsymbol{T}$ with probability $p_Z \in [0, 1]$. We distribute the effect of the $N_Z$ mixing variables equally across these targets, resulting in $0 \le i \le N_Z$ many disjoint sets $\boldsymbol{T}_i = \{X_j \mid \mathrm{La}_j = Z_i\}$. For example, we have $\boldsymbol{T}_1 = \{X_1, X_2\}$ and $\boldsymbol{T}_2 = \{X_4\}$ in Fig. 2.

To generate samples, we traverse $\boldsymbol{X}$ in topological order of the induced $\mathcal{G}$. For each $X_j$, we sample $\mathbf{B}_j = \{\boldsymbol{\beta}_{j1}, \ldots, \boldsymbol{\beta}_{jK_i}\}$ coefficient vectors, where $\boldsymbol{\beta}_{jk} \in \mathbb{R}^{|\mathbf{Pa}_j|}$ for all $k \in \{1, \ldots, K_i\}$ with $K_i = K$ if $X_j \in \boldsymbol{T}$ and $K_i = 1$ otherwise. We draw $\boldsymbol{\beta}_{jk} \in [-1, -\epsilon] \cup [\epsilon, 1]$ to avoid causal effects close to zero, and if possible also ensure that $|\boldsymbol{\beta}_{jk} - \boldsymbol{\beta}_{jk'}| > \epsilon$ for all pairs $k, k'$ to create sufficient class separation, where $\epsilon = 0.25$ by default. We then draw $S$ samples from a (mixture of) linear regression model(s) $(X_j | \mathbf{Pa}_j = \mathbf{y}) \sim \mathrm{MLR}\left(\mathbf{B}_j, \boldsymbol{\gamma}^j, \sigma^2\right)$, and standardize the resulting samples to generate an internally-standardized structural causal model (iSCM) [Ormaniec et al., 2024].

In the case studies, we consider a mixture of interventional datasets, as well as the flow cytometry dataset by Sachs et al., 2005 under the experimental setup in Wang et al., 2017. For both cases, we use the same scripts as in Kumar et al., 2024[2]. For the mixture of interventions, we have $N_Z = 1$ with $K = N_X + 1$ classes which defines a split into one observational and $K$ interventional datasets. Under a so-called diagonal or atomic setting, one node at a time undergoes intervention, resulting in disjoint sets $\boldsymbol{I}_k \subseteq \boldsymbol{X}$, here with hard interventions that fix $\boldsymbol{\beta}_{jk} = 0$ if $X_j \in \boldsymbol{I}_k$. A similar structure applies to the real-world dataset with 5846 samples and known manipulations on 5 of the 11 variables, namely $\boldsymbol{I}_1 = \{\mathrm{Akt}\}, \boldsymbol{I}_2 = \{\mathrm{PKC}\}, \boldsymbol{I}_3 = \{\mathrm{PIP2}\}, \boldsymbol{I}_4 = \{\mathrm{Mek}\}, \boldsymbol{I}_5 = \{\mathrm{PIP3}\}$.

**Evaluation Metrics**    To evaluate the discovered number of mixing components and assignments, standard metrics in clustering evaluation are appropriate, where we show the v-measure and the Adjusted Mutual Information (AMI) as two examples (e.g., Vinh et al., 2010). We average each score over $\boldsymbol{X}$ since we can associate each variable $X_j$ to a fixed assignment with $K_i$ components if $\mathrm{La}_j = Z_i$, else $K_i = 1$. To validate statements on *whether* observed variables are mixing targets,

---

[2]using the implementation of Mixture-UTIGSP at `https://github.com/BigBang0072/mixture_mec`

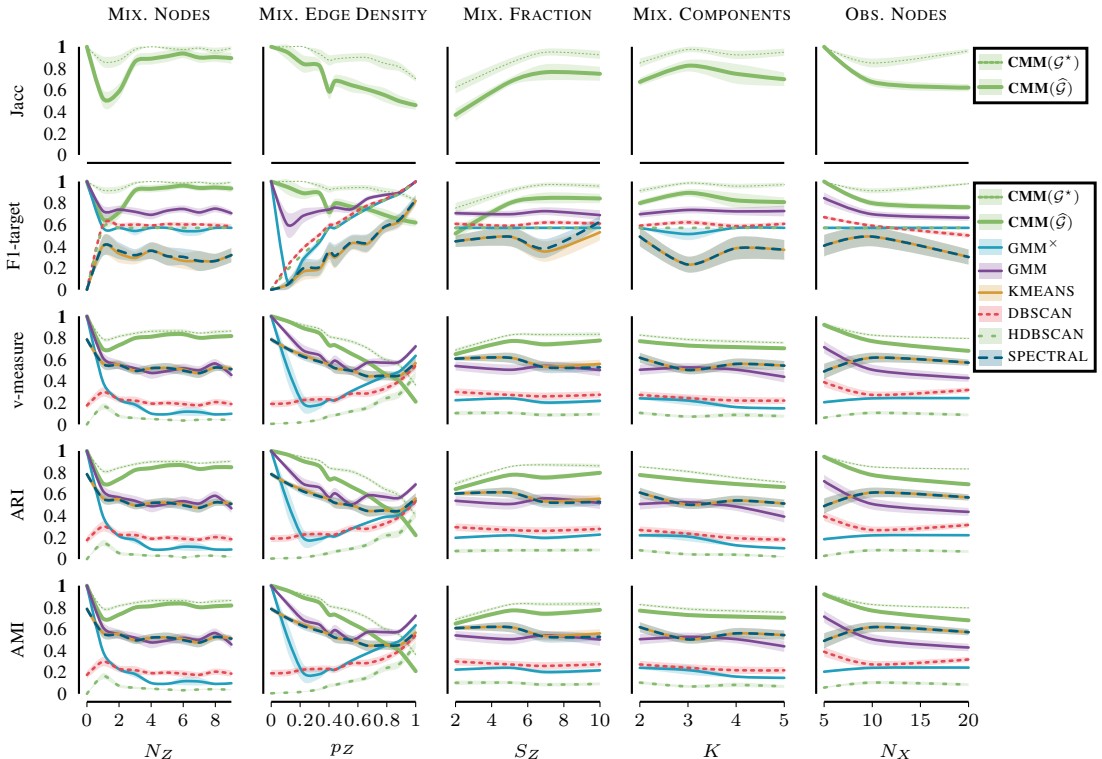

Figure D.1: *Discovering Mixing Structure.* In synthetically generated CMMs, we evaluate the quality of the recovered mixing structure, evaluating the affected observed variables (F1 (target)), variable sets affected by the same latent variable (Jacc) as well as per-node mixing assignments (AMI, ARI, v-measure).

we compute F1 scores (called F1-target) over the statement $X_j \in \boldsymbol{T}$ averaged over $\boldsymbol{X}$. To validate results on *which* mixing variables affect which mixing targets, we compute the Jaccard index (called Jacc) comparing the true sets $\{\boldsymbol{T}_1, \ldots, \boldsymbol{T}_{Z_m}\}$ to those returned by our algorithm.

We also compare the induced DAG $\mathcal{G}$ against the discovered DAG or CPDAG $\mathcal{G}'$. To give intuitive insight into correct vs. incorrect edge orientations, we show F1 scores over directed edge counts $E$ in $\mathcal{G}'$ (called F1-dir), where we note that in the case of CPDAGs, we only count edges that are directed with certainty. As this is a simplistic score mainly included for illustrative purposes, we also consider classical distance metrics. A common distance measure for two DAGs or CPDAGs $\mathcal{G}, \mathcal{G}'$ is the Structural Hamming Distance (SHD). More suitable for graphs $\mathcal{G}, \mathcal{G}'$ with a causal interpretation are the scores proposed in Wahl and Runge, 2024. The *s/c-metrics* (S/C) are based on counting separation statements and comparing their validity in $\mathcal{G}, \mathcal{G}'$. Scalable variants thereof are the *separation distances* (SD) that associate each pair of separable nodes in $\mathcal{G}'$ with a separation set $\mathcal{S}$ under a given separation strategy, and validate whether $\mathcal{S}$ remains separating in $\mathcal{G}$. We report the SC (without randomization) and the SD (with the 'parent' resp. 'pparent' type), which are defined both when the output is a DAG or a CPDAG[3].

**Baselines** As our method is the only one to discover the full $\mathcal{G}^Z$, we show (i) AMI and F1-target scores over $\boldsymbol{X}$ for all clustering baselines and wherever applicable for Mixture-UTIGSP, (ii) metrics on $\mathcal{G}$ for all causal discovery baselines, and (iii) Jaccard scores over $\{\boldsymbol{T}_i\}_i$ only for our method. We note that we apply all baselines without optimization of their hyperparameters using their implementations available in the `causal-learn`, `causal discovery toolbox (cdt)`, `causalDisco` and `dodiscover` Python libraries. For all conditional-independence tests, we use the Fisher-Z test given the linearity of our functional model. We ran the evaluations on an 11th Gen Intel Core i9 CPU.

---

[3]using the implementation of the metrics at `https://github.com/JonasChoice/sep_distances`

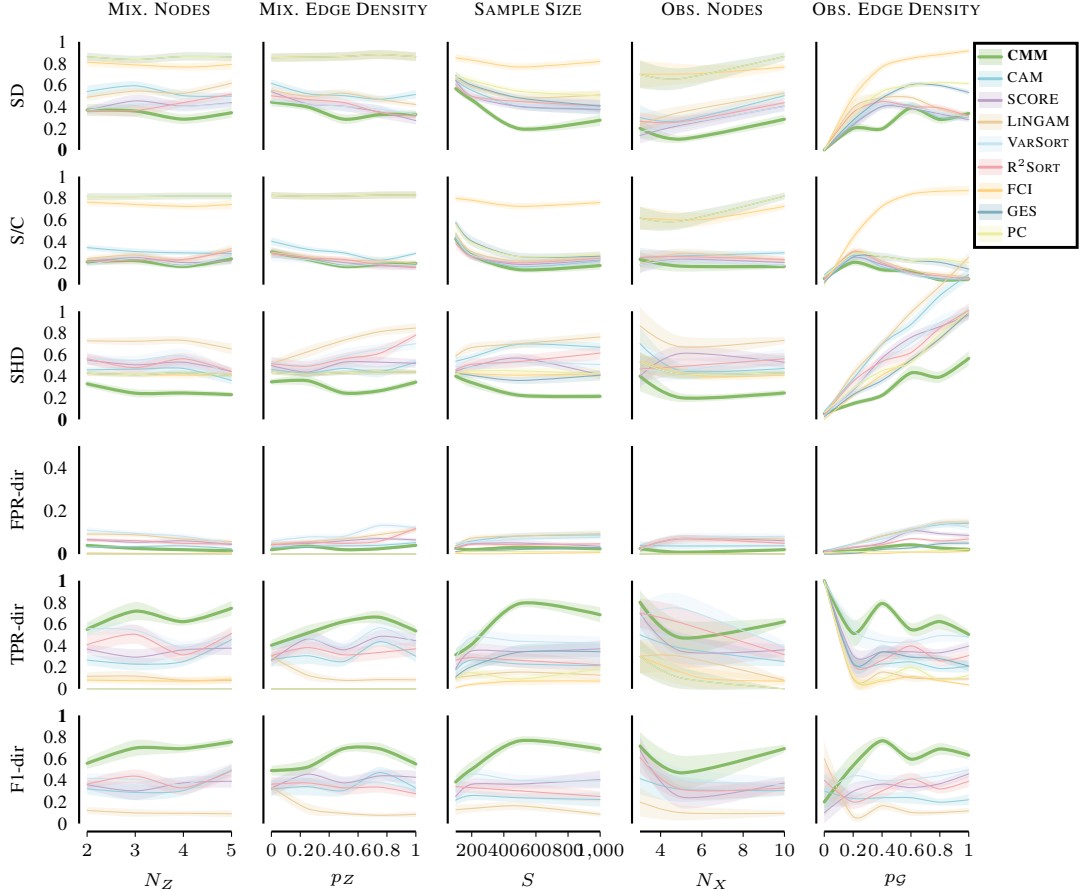

Figure D.2: *Discovering Causal Graphs.* In synthetically generated CMMs, we evaluate the quality of the causal DAG over the observed variables in terms of separation distances (SD, S/C), structural hamming distance (SHD) and directed edge counts (F1-dir, TPR-dir, FPR-dir).

**Discovering Mixing Structure** Fig. D.1 shows an extended variant of Fig. 4 in the main manuscript. The parameters are $N_X = 10, N_Z = 2, K = 2, p_Z = 0.4, p_\mathcal{G} = 0.4, S = 1000, S_Z = 5$, which are held fixed while changing one parameter of interest (columns in Fig. D.1), where we run $N_I = 10$ iterations for each parameter configuration. Different choices of the clustering algorithm besides the GMM, here shown in color for better readability, perform either worse on recovering targeted nodes (KMEANS, SPECTRAL) or mixing assignments (DBSCAN, HDBSCAN). We observe no noticeable differences between the clustering metric choices, so we report the AMI in the main manuscript.

**Discovering Causal Structure** Fig. D.2 shows an extended variant of Fig. 5 in the main manuscript. The parameters are $N_X = 10, N_Z = 4, K = 2, p_Z = 0.5, p_\mathcal{G} = 0.4, S = 500, S_Z = 5, N_I = 10$. All structural metrics (SD, S/C, and SHD) show stable performance of the CMM across the settings. The intuitive score TPR-dir furthermore suggests that our method performs well in distinguishing causal edge directions, improving as sample size $S$ increases, and as the likelihood of mixing $p_Z$ increases. In particular, the results suggest that "sparse" mixing $0 < p_Z < 1$ is most beneficial. We connect this to previous findings in the multi-environment setting [Perry et al., 2022] showing that identification of edge orientations is possible under the sparse mechanism shift hypothesis. This could inspire a future work direction generalizing this property from the multi-environment to the latent-mixed setting.

**Discovering Interventional Mixtures** Finally, Fig. D.2 extends Fig. 6 to show both mixing structure (Jacc, F1-iv) and causal structure (SD, S/C, SHD) discovery for the mixture of interventions. As a well-defined split into observational and interventional datasets exists for this setting, we also

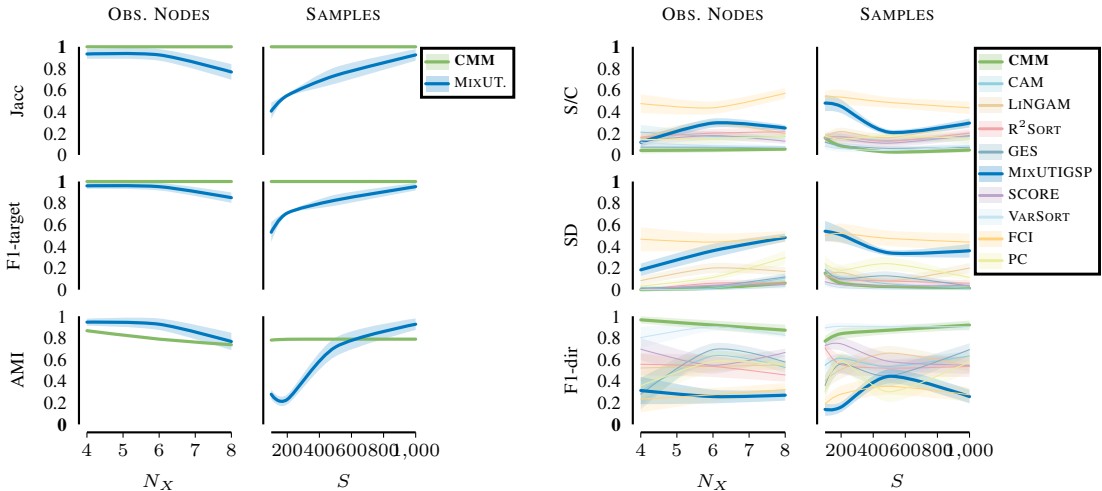

Figure D.3: *Mixtures of Interventions.* For data generated from an interventional mixture, we show the quality of discovered mixing (left) and graphs (right).

include Mixture-UTIGSP in the presentation (dark blue). The parameters $N_X \in \{4, 6, 8\}$ are as in Kumar et al., 2024, while we restrict the setting to up to $S = 1000$ samples, given that the true positive rates over $\mathcal{G}$ for the CMM (and VARSORT) already approach 1 for $S = 1000$; we refer to Kumar et al., 2024 to see the performance of Mixture-UTIGSP with more samples. Compared to Fig. D.1, the CMM performs much better on discovering whether ($X_j \in \boldsymbol{T}$) and which ($X_j \in \boldsymbol{T}_i$) variables are mixing targets. A likely reason for this is the fact that the hard interventions $\boldsymbol{\beta} = 0$ create a more distinct separation than re-sampling of $\boldsymbol{\beta}$ with $|\boldsymbol{\beta}_{jk} - \boldsymbol{\beta}_{jk'}| > \epsilon$.

# E Ablation Studies

As additions to our main experimental evaluation, we perform ablation studies on two questions.

(i) *Choice of Score-Based Algorithm.* We address the choice of the score-based causal discovery algorithm used together with the latent-aware BIC, showing GES alongside TOPIC.

(ii) *Effect of Latent Mixing Variables.* We take a closer look at how latent mixing affects causal discovery algorithms in practice. We hypothesize that spurious edges will appear between mixing targets, and study the extent to which the latent-aware BIC can prune these.

(iii) *Choice of Functional Forms.* Given that the results presented so far are restricted to linear settings, we test a proof-of-concept extension to nonlinear functional forms.

## E.1 Choice of Score-Based Algorithm

| | CAUSAL GRAPH | | |
|---|---|---|---|
| METRIC | **CMM** (TOPIC) | **CMM** (GES) | GES |
| SHD | $0.17 \pm 0.01$ | $0.31 \pm 0.02$ | $0.36 \pm 0.04$ |
| S/C | $0.13 \pm 0.03$ | $0.19 \pm 0.03$ | $0.26 \pm 0.03$ |
| SD | $0.17 \pm 0.02$ | $0.33 \pm 0.03$ | $0.48 \pm 0.04$ |

Table E.1: *Choice of Score-Based Algorithm.* We combine the CMM with different score-based algorithms (TOPIC, GES) as well as GES itself on causal discovery (cf. Fig. D.2).

While we used the latent-aware score $\mathrm{BIC}_{\hat{Z}}^{\mathrm{EM}}$ within the topological-ordering-based framework in the main evaluation, we can also use it within the GES algorithm, compared to GES with vanilla BIC. We compare these three variants in Table E.1 for the basic experimental parameters (as in

|  | MIXING ASSIGNMENTS (AMI) | | |
| FUNCTION $f$ | **CMM**-LINEAR | **CMM**-NATURAL-SPLINE | GMM |
| --- | --- | --- | --- |
| linear | $0.7624 \pm 0.0378$ | $\mathbf{0.7659 \pm 0.0296}$ | $0.3598 \pm 0.0431$ |
| quadratic | $0.6465 \pm 0.0935$ | $\mathbf{0.7163 \pm 0.0507}$ | $0.3116 \pm 0.0452$ |
| cubic | $0.5984 \pm 0.0893$ | $\mathbf{0.6876 \pm 0.0589}$ | $0.2841 \pm 0.0554$ |
| exp | $0.6345 \pm 0.0860$ | $\mathbf{0.7025 \pm 0.0544}$ | $0.3007 \pm 0.0604$ |
| log | $0.6628 \pm 0.0849$ | $\mathbf{0.7100 \pm 0.0527}$ | $0.3100 \pm 0.0615$ |
| sin | $0.6452 \pm 0.0884$ | $\mathbf{0.7035 \pm 0.0561}$ | $0.3144 \pm 0.0603$ |

Table E.2: *Choice of Functional Forms.* Under different generating processes $f$ for the synthetic data, we compare the linear and a nonlinear instantiation of our model, in bold to indicate the best per-row model.

Fig. D.2). Regarding the choice of the algorithm, the topological-ordering-based variant (**CMM** (TOPIC)) appears to have better practical performance in our experiments. This experiment also confirms that regarding the score, $\mathrm{BIC}^{\mathrm{EM}}_{\hat{Z}}$ with MLR fitting (**CMM** (GES)) provides a benefit over plain BIC (GES).

## E.2 Effect of Latent Mixing Variables

We are also interested in how exactly the presence of latent mixing variables influences the graphs $\mathcal{G}'$ returned by causal discovery methods unaware of mixing. As the latents $Z$ introduce dependencies between the mixing targets $T$, we presume that the reported $\mathcal{G}'$ will contain additional spurious (FP) edges.

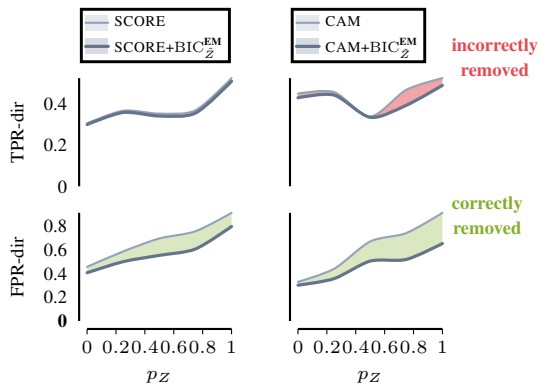

Figure E.1: *Effect of Latent Mixing.*

In Fig. E.1 (light gray) we observe that the FPR in $\mathcal{G}'$ indeed increases as the probability $p_Z$ of $X_j \in T$ increases (shown for SCORE and CAM). Given this trend, we investigate whether we can correct the result by pruning any FPs that arise from mixing. Thus, we apply the CMM to each node $X_j$ given its causes in $\mathcal{G}'$, fit an MLR, and use the $\mathrm{BIC}^{\mathrm{EM}}_{\hat{Z}}$ to remove any redundant parents of $X_j$ under this model. This results in a graph $\mathcal{G}''$, also shown in Fig. E.1 (dark gray).

The shaded regions indicate the extent to which FP edges are removed correctly (green), respectively, TP edges are removed incorrectly (red). The $\mathrm{BIC}^{\mathrm{EM}}_{\hat{Z}}$ prunes some of the spurious and almost no causal edges. However, there still remain additional FPs in $\mathcal{G}''$ when $p_Z$ increases. This is perhaps due to practical limitations of EM in estimating the correct mixing, leaving room for future improvements.

## E.3 Choice of Functional Forms

While we consider it reasonable to limit the scope of our work to linear models, we also conducted a proof-of-concept case study to demonstrate the integration of nonlinear models.

For this, we replace the linear regression mixtures with nonlinear variants, specifically, a natural spline. We again use the EM algorithm to infer the mixture components and the BIC criterion to pick their number. Table E.2 shows how well we can reconstruct the mixture components (cf. Fig. 4 in the main text), measured by Adjusted Mutual Information (AMI) averaged over each node in 10-node graphs. We show the CMM with the MLR in as our main presentation (left) and the one with a mixture of natural splines (right) under different true generating functions (rows). The experimental setting corresponds to the same base parameters that we base Fig. 4 on.

The results match the expectations, where the models perform very closely in the linear case, but the MLR degrades in performance under nonlinearity. Replacing it with the mixture of splines allows covering the nonlinear cases reasonably well.

**Conclusions**    We reach a similar conclusion from our questions in Sections E.1 and E.2: using the latent-aware BIC, we can expect not a substantial, but at least some improvement in causal discovery – be it via correcting the outputs of causal discovery algorithms (E.2), as an scoring criterion in GES replacing vanilla BIC (E.1), or similarly within TOPIC (Fig. D.2) – while in addition being able to discover the latent structure $Z$ that can point us to subsamples of the data with a distinct causal generating process. Finally, the proof-of-concept experiment using nonlinear functional forms in Section E.3 also supports exploring our algorithm with more flexible regression mixtures.

# F    Limitations

One limitation regards the assumption of oracle MLR parameters; as a result, Alg. 1 is only consistent under appropriate conditions, such as a good-enough EM initialization regime, and otherwise should be viewed as a reasonable approximation to the problem. Studying the consistency of the estimates derived by the EM algorithm is a difficult problem, and the literature is limited to rather strict scenarios, which is why we cannot easily provide strict guarantees.

Given this, we can still note that the experimental evidence supports the connection we make in Conjecture 3.5. There is also a reasonable mechanism that could theoretically explain the observed behaviour (asymptotically). Simply put, when the data truly follow the MLR model, it is more likely for our EM implementation to align with the oracle; when this does not happen, it is likely due to a faulty model, in which case our downstream process would anyways (asymptotically) reject the specific MLR hypothesis in favour of the correct alternative.

Furthermore, we limit the scope of the exposition mainly to linear additive noise models. While our guarantees for the latent-aware oracle BIC can in principle be transferred to a nonlinear setting, this would however require access to oracle mixture parameters, and the connection to estimates obtained using EM is less clear under nonlinearity. This point, while we supplement it with a small empirical analysis, requires a deeper theoretical analysis, which we postpone to future work.

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
