# OpenReview forum: "Causal Mixture Models: Characterization and Discovery"
_NeurIPS.cc/2025/Conference — NeurIPS 2025 poster_

### Official Review · Reviewer_PSSz · 2025-06-24

**Clarity:** 2
**Significance:** 3
**Originality:** 2
**Rating:** 5
**Confidence:** 3

**Summary:**

The authors study the problem of causal discovery using data from different distributions/environments which they frame as a causal mixture model using potentially multiple discrete mixing variables. These mixing variables are assumed to be unobserved. They propose a score-based EM-style method using BIC that jointly finds the causal ordering and the mixing variables. They provide a theoretical analysis of their approach as well as numerical experiments.

**Questions:**

- In the motivational example you provided (with the antibiotics), it is not a setting with multiple independent latent mixing variables, right? Can you give a real-world example where there are such independent variables? I think this would strengthen the motivation for this work.
- In the description of the algorithm, what do you mean by the edge $Pa_j \rightarrow X_j$? How does it relate to the formula in line 206? Do you mean the edge $X \rightarrow Y$?

**Ethical Concerns:**

["NO or VERY MINOR ethics concerns only"]

**Final Justification:**

The authors detailed various ways to improve the clarity of the paper during the rebuttal.

**Limitations:**

There is no specific section addressing limitations. Some limitations are mentioned in the conclusion.

**Paper Formatting Concerns:**

No.

**Quality:**

3

**Strengths And Weaknesses:**

**Strengths:**
- The authors study the relevant problem of causal discovery with data from different environments where the environment-assignment is unknown
- They provide extensive experiments on simulated as well as real-world data
- The paper is structured clearly, and easy to follow, but some of the mathematical notation can be streamlined (see points in weaknesses).

**Weaknesses:**
- The method rests on the strong assumption of linearity as recognized by the authors. The authors claim in the conclusion that the method can "easily incorporate richer models", I think it would have strengthened the contribution to explore this in some way, e.g. in the experiments.
- There seems to be a strand of work missing in the related work section, namely methods that try to detect changing contexts/regimes, e.g. R-PCMCI [1].
- The contribution of framing the problem in terms of a mixture model with multiple latent variables seems somewhat minor to me, also the proposed method seems straightforward for the problem and rests heavily on existing methods such as GES, EM, and TOPIC. This limits the novelty a bit, however I recognize that the application to the described causal discovery problem is novel.
- In my opinion the introduction of the causal model in section 2 can be streamlined a bit. For example, I think it would help to express $\beta$ as a function of the latent mixing variable (instead of allowing for general influence in the mechanism $f$ of $X_j$). This would show that this is the only point of influence for the latent. I also find it a bit confusing that you distinguish between the parent set of $X_j$ and the latent $La_j$: Isn't the mixing variable also a parent of some of the $X_j$s? Or do you define the parent set $Pa_j$ in line 88 as a subset of the observed variables?
- Some of the concepts that not all readers are familiar with could be recapped/introduced in the main text, e.g. Adjusted Mutual Information, furthermore the notation $BIC_{\hat{Z}}$ could be introduced more explicitly.
- Typos:
  - line 64 "be extended the mixed setting"
  - line 80 resp.
  - line 127 "a frameworks"
  - line 180 distinguish instead of identify?

---

> ### Author Rebuttal · Authors · 2025-07-31
>
> Thank you for your extensive response; we truly appreciate your time and constructive feedback.
>
> Please consider our responses to your comments below.
>
> # Response to General Comments
>
> ## Presentation
>
> * In the related work, we will include a reference to time series methods such as RPCMCI. These methods use EM to discover regimes along the temporal dimension and, therefore, additionally rely on the temporal ordering of the data.
>  * We also value the suggestions for streamlining our formulation of the causal model; we agree with your viewpoint and will express $\beta$ to be a function of
> the respective mixing variable. We also note that you are correct that $Z$ are themselves causal parents, but we write $\text{Pa}_i$ to distinguish the observed from the unobserved parents.
> * We appreciate the effort taken to point out our typos, we have corrected them in the current manuscript.
>
>
> ## Nonlinear Models
>
> While we consider focusing our analysis on linear models useful on its own, we would like to justify our claim on the "easy" extensibility of our framework to nonlinear models. We have therefore we conducted a small proof-of-concept study to support this claim.\
> Please consider our results, below.
>
> In the following, we replace the linear-regression mixtures with nonlinear variants, here a natural spline. We again use the EM algorithm to infer the mixture components and the BIC criterion to pick their number.
> The following table shows how well we can reconstruct the mixture components  (Fig. 4 in the manuscript), measured by Adjusted Mutual Information (AMI) averaged over each node in 10-node graphs.\
> We show the CMM with the MLR in as our main presentation (left) and the one with a mixture of natural splines (right) under different true generating functions (rows).
>
> | _AMI_           | **CMM-linear**     | **CMM-natural-spline** | **GMM** (baseline)            |
> |-----------------|--------------------|-------------------------|---------------------|
> | *f*: linear     | **0.7624** ± 0.0378    | **0.7659** ± 0.0296         | 0.3598 ± 0.0431     |
> | *f*: quadratic  | 0.6465 ± 0.0935    | **0.7163** ± 0.0507         | 0.3116 ± 0.0452     |
> | *f*: cubic      | 0.5984 ± 0.0893    | **0.6876** ± 0.0589         | 0.2841 ± 0.0554     |
> | *f*: exp        | 0.6345 ± 0.0860    | **0.7025** ± 0.0544         | 0.3007 ± 0.0604     |
> | *f*: log        | 0.6628 ± 0.0849    | **0.7100** ± 0.0527         | 0.3100 ± 0.0615     |
> | *f*: sin        | 0.6452 ± 0.0884    | **0.7035** ± 0.0561         | 0.3144 ± 0.0603     |
>
> The results match our expectations, where the models perform very closely in the linear case, but the MLR degrades in performance under non-linearity. Replacing it with the mixture of splines allows covering the non-linear cases reasonably well.
>
> This proof-of-concept experiment supports exploring our algorithm with more flexible regression mixtures, and we can include it as an ablation study to encourage this. Indeed, our guarantees for the latent-aware oracle BIC can be transferred to a nonlinear setting. However, this would require access to oracle mixture parameters, and the connection to estimates obtained using EM is less clear under non-linearity. This point would require a deeper analysis in future work, but we hope that the reviewers agree that the parametric case is of sufficient contribution for the scope of this work.
>
>
> ### Implementation Details
> As we cannot provide source code here, we summarize the changes we made to the implementation. The source code included in the supplement requires the following small changes,
>
> 1. using a different regression-mixture within our algorithm. This can be done in ``src/mixtures/mixing/mixing.py`` by changing the syntax of the regression formula passed to the R-script.
> 2.  using a different data-generating function $f$ for each causal relationship in the randomly generated DAGs. Implementation support for this already exists under ``src/exp/gen/synthetic/data_gen_mixing.py`` where the linear function used in each cause-effect relationship can be replaced by other functional forms. To visualize these functional forms, one can refer to the notebook ``src/examples/demo_mixing.ipynb`` and modify the parameter ``F``, such as  ``params['F'] = FunType.CUB`` for cubic functions.
>
> The experimental setting corresponds to the same base parameters that we used in main manuscript.
>
> # Response to Questions
>
> ### Motivating Example
>
> This is a good point. We can indeed extend our motivational example to motivate multiple latent mixing variables. For example, independent latent variables could correspond to (1) unknown batch effects due to different laboratories from which measurements come and (2) prior exposure related to demographic location. Following your suggestion, we improved the motivational premise in the updated manuscript as follows.
>
>    > Take for example a country-wide study of anti-microbial resistance in hospitalised patients focused on a resistant pathogen such as Methycillin-Resistant Staphylococcus Aureus (MRSA) [4]. For each patient, phenotypic susceptibility to multiple antibiotics is measured; for each antibiotic, each medical center may choose one of the collaborating laboratories to perform each test, thereby introducing a latent variable corresponding to a batch effect, each independent from the others.\
>   Further, each patient's medical history includes prior exposure to pathogens with known cross-resistance to MRSA, such as Enteroccocus [5]. The plasmid profile of the latter in each geographical region largely affects its susceptibility, say to Vancomycin [2], which in turn affects the cross-resistance of MRSA [1]. Despite being well documented, this variable is not routinely measured by medical centers, and is hence an independent latent variable that defines the mechanism under which the presence of Enteroccocus affects the MRSA cross-resistance [3].\
>    Such settings, where multiple observed variables are each affected by one independent latent factor, are common in practice. Yet, this premise is not currently addressed by standard causal inference methods, which often rely on assumptions like causal sufficiency or no unmeasured confounders. Both of these assumptions are violated in this setting, often leading to the discovery of spurious dependencies, or to missing true causal relations, altogether  [6, 7].
>
>     * [1] Arredondo-Alonso S, Top J, McNally A, Puranen S, Pesonen M, Pensar J, Marttinen P, Braat JC, Rogers MRC, van Schaik W, Kaski S, Willems RJL, Corander J, Schürch AC. Plasmids Shaped the Recent Emergence of the Major Nosocomial Pathogen Enterococcus faecium. mBio. 2020 Feb 11;11(1):e03284-19. doi: 10.1128/mBio.03284-19. PMID: 32047136; PMCID: PMC7018651.
>
>     * [2] Boumasmoud M, Dengler Haunreiter V, Schweizer TA, Meyer L, Chakrakodi B, Schreiber PW, Seidl K, Kühnert D, Kouyos RD, Zinkernagel AS. Genomic Surveillance of Vancomycin-Resistant Enterococcus faecium Reveals Spread of a Linear Plasmid Conferring a Nutrient Utilization Advantage. mBio. 2022 Apr 26;13(2):e0377121. doi: 10.1128/mbio.03771-21. Epub 2022 Mar 28. PMID: 35343787; PMCID: PMC9040824.
>
>     * [3] Cong Y, Yang S, Rao X. Vancomycin resistant Staphylococcus aureus infections: A review of case updating and clinical features. J Adv Res. 2019 Oct 12;21:169-176. doi: 10.1016/j.jare.2019.10.005. PMID: 32071785; PMCID: PMC7015472.
>
>     * [4] Hasanpour AH, Sepidarkish M, Mollalo A, Ardekani A, Almukhtar M, Mechaal A, Hosseini SR, Bayani M, Javanian M, Rostami A. The global prevalence of methicillin-resistant Staphylococcus aureus colonization in residents of elderly care centers: a systematic review and meta-analysis. Antimicrob Resist Infect Control. 2023 Jan 29;12(1):4. doi: 10.1186/s13756-023-01210-6. PMID: 36709300; PMCID: PMC9884412.
>
>     * [5] Li G, Walker MJ, De Oliveira DMP. Vancomycin Resistance in Enterococcus and Staphylococcus aureus. Microorganisms. 2022 Dec 21;11(1):24. doi: 10.3390/microorganisms11010024. PMID: 36677316; PMCID: PMC9866002.
>
>     * [6] Huang B, Zhang K, Zhang J, Ramsey J, Sanchez‑Romero R, Glymour C, Scholkopf B. Causal Discovery from Heterogeneous/Nonstationary Data: Skeleton Estimation and Orientation Determination. Journal of Machine Learning Research. 2020 May;21(89):1–53.
>
>     * [7]  Mazaheri B, Squires C, Uhler C. Synthetic Potential Outcomes and Causal Mixture Identifiability. In: Proceedings of the 28th International Conference on Artificial Intelligence and Statistics (AISTATS 2025); 2025 May; Mai Khao, Thailand. PMLR, vol. 258.
>
> ### Algorithm Notation
>
> In the description of the algorithm, we will refer to $\text{Pa(Y)}$ instead of $\text{Pa}_j$, thank you for pointing out this inconsistency. To clarify, we refer indeed to the edge $X \rightarrow Y$ that we consider including in the current parent set $\text{pa}(Y)$ for $Y$, for which we fit mixture models under these given parent sets.
>
>
> We are happy to address any additional points that may arise.

---

> > ### Comment · Reviewer_PSSz · 2025-08-04
> > **Response to authors**
> >
> > Dear authors,
> > thank you for your response, which has addressed most of my questions. However, I find it hard to judge how your proposed changes to the presentation (especially in section 2) will be implemented in the final version, and how this will affect the overall clarity of the theoretical part of the paper. Maybe you could provide more detail on this?

---

> ### Author Response · Authors · 2025-08-05
> **Proposed Changes to the Presentation**
>
> Dear reviewer, thank you very much for your response.
>
> We propose the following mild changes to Section 2, and welcome any suggestions.
>
> > To formalise this, we also consider a set of discrete, unobserved random variables $\boldsymbol{Z} = ${$Z_1,\ldots,Z_m$}, with $m \leq n$, each following a categorical distribution $Z_i \sim \mathrm{Categorical}(\boldsymbol{\gamma}^i)$ with $Z_i \in \{1, \ldots, K_i\}$. That is, each $\boldsymbol{\gamma}^i$ lies on a $K_i$-dimensional probability simplex $\boldsymbol{\gamma}^i = (\gamma^i_1, \ldots, \gamma^i_{K_i})$ with $\sum_{k=1}^{K_i} \gamma^i_k = 1$, so that $\mathbb{P}(Z_i = k) = \gamma^i_k$. We call these random variables *mixing variables*.
>
> > The causal mechanism of each observed random variable in $\boldsymbol{X}$ depends on a set of observed causal parents as well as exactly one of the unobserved, latent $\boldsymbol{Z}$, as determined by the surjective map $\mathrm{La} : \boldsymbol{X} \to \boldsymbol{Z} : X_j \mapsto Z_i$, in which case we simply write $\mathrm{La}_j = Z_i$. The mixing variable directly affects the parameters of the causal mechanism, which we therefore express as a function $b_j: [K_i] \to \mathbb R^{|\mathbf{Pa}_j|}: k \mapsto \beta^{jk}$  mapping each value $k$ of $Z_i$ to a linear coefficient vector $\beta^{jk}  \in \mathbb R ^{|\mathbf{Pa}_j|}$. Hence, the parameters of the functional dependency $f$ are the collection of vectors $B_j=(\beta^{j1}, ..., \beta^{j K_i}$); this consists of one coefficient vector $\beta^{jk}$ for each mixing coefficient $1\leq k \leq K_i$, and each such vector has a dimension equal to the number of parents $\mathbf{Pa}_j$ of $X_j$.
>
>
> > Summarizing, we can now model each random variable $X_j$ as generated from its observed causes $\mathbf{Pa}_j \subseteq \boldsymbol X$ by the causal function $f$ and the coefficients $b_j$ that depend on the latent $Z_i = \mathrm{La}_j \in \boldsymbol{Z}$, where we recall that $\boldsymbol Z \cap \boldsymbol X = \emptyset.$ Then, we have
>
> >  $X_j = f(\mathbf{Pa}_j,  b_j)+ N_j  \qquad \text{ with }\qquad  f( x , b_j (k)) =$  $\beta^{jk \top}$ $  x   \qquad   \qquad (1), $
>
> >  where $N_j  \perp\kern-5pt \perp \mathbf{Pa}_j$ is additive Gaussian noise, $N_j \sim \mathcal{N}(0, \sigma^2)$.
>
> The main changes follow your suggestions, where we
> - explicitly write $\mathbf{Pa}_j \subseteq \boldsymbol X$ to refer to the observed causal parents while $\mathrm{La}_j$ is a single unobserved parent, and
> - express the linear coefficients as a function $b_j$ of $\mathrm{La}_j$, so that the dependence on the latents enters directly (and only) through $b_j$  instead of appearing as a general argument of $f$; see Eq (1).
> ---
> *Note: There seems to be an issue with LaTeX code display where subscripts of two or more letters, as in ``\beta_{jk}``, result in display errors when used in longer equations.  Since the same works fine for superscripts, ``\beta^{jk}``:  $\beta^{jk}$, we temporarily moved the indices for all $\beta$ to the superscript in our response above. In the manuscript, we will keep using subscripts as in the submitted version.*

---

> > ### Comment · Reviewer_PSSz · 2025-08-07
> > **Response**
> >
> > Thank you for your detailed response. I think that the proposed changes will improve the overall clarity. I have no further questions and I will raise my score.

---

### Official Review · Reviewer_Vdb6 · 2025-06-28

**Clarity:** 4
**Significance:** 3
**Originality:** 4
**Rating:** 4
**Confidence:** 3

**Summary:**

This paper addresses statistical causal discovery under the assumption of unobserved subgroups. While prior methods have incorporated such assumptions, they typically rely on a single latent environment variable that affects all observed variables. In contrast, this work proposes a more realistic setting in which multiple latent mixing variables independently influence distinct subsets of observed variables. The authors present a novel method based on this assumption and demonstrate that it can be implemented as an extension of score-based causal discovery, specifically Greedy Equivalence Search (GES), using the Bayesian Information Criterion (BIC).

They show that while causal discovery is known to be non-identifiable under simple linear additive Gaussian noise models, the introduction of a mixture structure improves identifiability. The paper further proposes the use of the Expectation-Maximization (EM) algorithm for likelihood estimation within the mixture model, and argues, both intuitively and theoretically, that EM yields sufficient identifiability in practice.

Experiments on simulated data demonstrate that the proposed method outperforms existing approaches in both recovering the mixture structure and the underlying causal graph. Additionally, evaluations on real-world cell signaling data show that the method achieves strong causal structure recovery performance.

**Questions:**

1. Could the authors provide a discussion on how the inferred mixing variables in the Sachs dataset relate to known experimental conditions or biological states?

2. Could the authors comment on possible directions for more general research into the interpretability of inferred mixing variables?

**Ethical Concerns:**

["NO or VERY MINOR ethics concerns only"]

**Final Justification:**

The rebuttal provided a detailed explanation of the Sachs dataset results, convincingly showing that the inferred mixing variables can be meaningfully related to known experimental conditions and biological states. For the question on interpretability, the authors proposed plausible future directions (e.g., LIME, Shapley values), but these remain potential approaches without concrete implementation or results. Given this, I lean toward recommending acceptance, but will keep my original rating unchanged.

**Limitations:**

As mentioned above, there is room for improvement in the discussion of the interpretability of the inferred mixing variables.

**Paper Formatting Concerns:**

There are no particular paper formatting concerns.

**Quality:**

4

**Strengths And Weaknesses:**

Strengths:
This paper presents a causal model based on a more realistic assumption in which multiple latent mixing variables independently influence different observed variables. The authors make a valuable theoretical contribution by demonstrating that this formulation can be addressed using an extension of score-based causal discovery with the Bayesian Information Criterion (BIC). This significantly enhances the practical applicability of statistical causal discovery methods. Moreover, the proposed method incorporates a practical inference procedure using the Expectation-Maximization (EM) algorithm, and the authors provide theoretical justification suggesting that EM yields sufficient identifiability in practice. The effectiveness of the proposed approach is convincingly demonstrated through both synthetic and real-world data.

Weaknesses:
While the assumption that each observed variable may be influenced by a distinct latent mixing variable is realistic and well-supported by both theoretical analysis and empirical evaluation, it comes at the cost of producing highly complex inferred causal structures. As a result, interpreting the estimated mixing variables in a meaningful and actionable way becomes challenging. In the context of causal inference, even when the structure is correctly recovered, models that lack interpretability tend to have limited value in practical applications. Addressing this issue, either by proposing a strategy for interpreting the inferred latent variables or by providing concrete examples from real-world datasets (e.g., the Sachs dataset) where the inferred mixing variables align with known experimental conditions or biological states, could greatly enhance the credibility and practical utility of the proposed method.

---

> ### Author Rebuttal · Authors · 2025-07-31
>
> We thank you for your time and thought-provoking comments.
>
> Please consider our response to your questions below.
>
> # Response to Questions
>
> **1. Experimental States in Sachs et al.**
>
> Yes, there  is a known correspondence of the mixing variables to experimental conditions for this data.
>
> The dataset arises from single-cell protein-signalling data of the human immune response system, wherein each of the $11$ observed variables corresponds to the activity of one compound of interest: either a protein or a phospholipid. The dataset contains (among others) 5 experimental conditions, in each of which a particular molecular modifier has been applied to the cells, such that the activity of exactly one of the $11$ compounds of interest is affected. This results in a known change of the measured activity for the corresponding compound.
>
> In our experiments, we combine the data from all experimental conditions, and we do not make their origin available to the algorithm. Hence, in the pooled data, each variable is affected by a single latent one. The usefulness and relative prevalence of this dataset as a real-world case study among causal inference methods arises from both
>
>  1. the well-studied underlying causal mechanisms, and, in our case, also
>  2. from the known ground-truth values of the resulting latent variables.
>
> For example, considering the node "`Akt`", there are two mixture components: one is the experimental condition where the so-called Akt inhibitor was applied, inhibiting the activation of Akt directly; the other mixture component comprises the remaining samples where Akt is in its baseline condition. Similar interventions have been applied to each of the remaining $4$ variables shown in Table 1 (PKC, PIP2, Mek, and PIP3) which have been intervened upon in a different experimental condition.
>
> We hope that this explanation makes the setting more clear; we will expand appropriately the description of the Sachs dataset in the improved manuscript.
>
>
>
> **2. Interpretability**
>
> We agree that interpretability is a valuable property of any causal model. Like many statistical models, the full causal structural model may contain information that can overwhelm a human researcher, more so when it includes latent variables. Below we give some pointers on how to interpret their meaning.
>
> * *Interpretation of mixing variables:*\
> The categorical nature of the mixing variables has the benefit that a domain expert can directly inspect the components and their number. For example, someone interested in a specific target node $Y$ with cause $X$ and affected by a latent $Z$ can readily visualize the $k$ components of $Z$ as "clusters" and inspect how the causal mechanism $X\rightarrow Y$ differs in the respective groups, similar to in Fig. 1 of the manuscript.
> \
>  When  $Y$ has more than one cause, a pair-wise analysis or a dimensionality reduction technique could allow for a similar visual analysis. It is also interesting to see which nodes $Z$ jointly influence a subset of nodes, as this is a strong indication of latent subgroups influencing a part of the system. When sufficient background domain knowledge is available, one could match the $k$ components to known genetic subgroups or biological states. If no such expert knowledge is available, one might gain novel insights into the existence of possible causal mechanisms that were previously unknown.
>
> * *Follow-up work:*\
> Your question inspires some ideas for future work where we further interpret the meaning of a given $Z$, especially when they point to previously unknown states.  To this end, one could explore explainability approaches such as LIME or Shapley values. For this, one could see the latent mixing variables $Z$ as classification targets, and then test whether any known or background knowledge features are important predictors of $Z$ using LIME or Shap. For example, one might find whether the latent components arise from a population group of a certain age, demographic, or other background.
>
> Please let us know if we can address any remaining questions or concerns.

---

### Official Review · Reviewer_qK58 · 2025-07-02

**Clarity:** 2
**Significance:** 3
**Originality:** 3
**Rating:** 4
**Confidence:** 3

**Summary:**

This paper considers the problem of causal discovery in linear Gaussian structural equation models (SEMs) when the data consist of multiple subpopulations. Specifically, unlike previous work where data can typically be partitioned into multiple domains or environments, all observed data here are collected within a single environment. The authors first define the causal mixture model (CMM), where unobserved mixing variables $Z_i$ are introduced to index different linear coefficients or weights. Subsequently, they present identification results and algorithms. The authors show that, given an oracle maximum likelihood estimator, the structure can be uniquely identified using an adapted Bayesian Information Criterion (BIC) score. Furthermore, they propose an algorithm based on greedy equivalence search that employs the Expectation-Maximization (EM) algorithm to compute the BIC score. The effectiveness of the proposed method is demonstrated on both synthetic and real datasets.

**Questions:**

1. In Figure 2, what does the edge Z_1 -> X_1 mean? According to Equation (1), the mixing variable is introduced to distinguish the linear coefficients. Since $X_1$ is a source variable, I am not sure how different values of $Z_1$ would affect $X_1$.
2. Do the identification results and the proposed algorithm rely on the prior assumption that all observed variables have equal noise variance? It is known that while linear additive Gaussian noise models are generally not identifiable, linear Gaussian models with equal noise variance are identifiable and can be identified using maximum likelihood estimation.
3. Regarding Theorem 3.4:

    (a) What does $Y$ represent? According to the notation in Section 2, $Y$ should be the parents of $X$. If it represents some type of "outcome variable", then I am unclear about the meaning of the conditional distribution $\boldsymbol{X}|Y$.

   (b) In line 179, I recommend that the authors avoid using the term “Markov equivalence class” if the intent is to state that the models can be distinguished based on the observational distribution.

    (c) Is Theorem B.2 the same as Theorem 3.4? If so, then it seems that Lemma 3.3 is derived from Theorem B.2 and should be considered as a corollary.

4. In lines 210–212, how is the BIC score computed? Does it account for the unknown number of mixtures by employing the EM algorithm, or does it follow the conventional BIC formulation that assumes a single mixture component?

**Ethical Concerns:**

["NO or VERY MINOR ethics concerns only"]

**Final Justification:**

I recommend the paper for acceptance, but would like to maintain my current evaluation.

While the overall quality of the paper is good, there is still room to improve the clarity and presentation.

**Limitations:**

The authors discuss the limitations of the proposed model and identification algorithm.

**Quality:**

3

**Strengths And Weaknesses:**

**Strengths**:
1. The presentation of the paper is clear and easy to follow, although adding more explanations and examples may further aid in understanding the technical details.
2. The problem is well-motivated and clearly defined.
3. The authors provide a clear description of the background on greedy equivalence search (GES) and clearly state the limitations and conjectures.

**Weakness**:
The presentation of certain technical details and theoretical results (e.g., Theorem 3.4) is difficult to understand (see the questions below).

---

> ### Author Rebuttal · Authors · 2025-07-31
>
> We appreciate your time and thank you for the detailed feedback.
>
> Please consider our responses to your points below.
>
> # Response to Questions
>
> **1. Source Variables**
>
> This is a good question; we will clarify the meaning of edges such as  $Z_1 \rightarrow X_1$ (Fig. 2), whenever $X_1$ is a source node of the causal graph.\
> Note that the GMM used in the sources can be seen as a special case of the MLR model if we use an implicit input variable with a constant value of 1. Indeed, re-visiting Eq. (4) we would then have $\mathbf Y= \{\mathbb 1\}$, where $\mathbb 1$ is a deterministic input variable $\mathbb 1 = 1$.\
> Then, we would obtain the GMM as
> $$ p_{X|\{\mathbb 1\}}^{\mathrm{MLR}}(x;\mathbf\beta,\mathbf\gamma,\sigma^2)=\sum_{k=1}^K\gamma_k p^{\mathcal N}_{X}(x;\beta_k\cdot 1,\sigma^2),$$
> which the reader can easily verify to be the p.d.f. of the GMM model.
>
> This also gives a seamless approach: We use EM and the BIC to infer latent mixing for both sources and non-sources, where we fit a GMM in the case of sources and an MLR otherwise.
>
> We will include this explanation after Eq. (4) to be clear on its meaning for source nodes.
>
> **2. Equal Noise Variances and Identification**
>
> Indeed, methods exist to identify linear additive Gaussian noise models with equal variance (under further conditions, e.g., non-perfect dependencies), and what is more, also when other constraints are added (e.g., that the coefficients are all less than the unit).
> Our identification results do not rely on such assumptions. Instead, we base identification on the effects of the latent mixing variables and the asymmetries they introduce in the causal directions.
> Thus, our results can be seen as orthogonal to the approach of restricting the functional model with further assumptions, and can therefore address scenarios where these perhaps unrealistic assumptions do not hold.
>
>
> **3. Formulation of Theorem 3.4**
>
> * **(a)** We thank you for spotting this typesetting error. Here the correct distribution is $X|\mathbf Y$, in alignment with the notation of Eq. (4).\
> We have made the correction $\mathbf X\to X$ and $Y\to \mathbf Y$ in the improved manuscript.\
> The intended meaning is as you correctly assumed.
>
> * **(b)** We thank you for and agree with this recommendation. Following your suggestion, we refrain from using 'Markov Equivalence Class' here to avoid confusion.
>
> * **(c)** Indeed, the two are the same. We have corrected the theorem numbering in the updated appendix.\
>   As for Lemma 3.3, we agree and will make the appropriate change.
>
> **4. Specifying the Number of Mixtures**
>
> Exactly, the BIC in lines 210-212 accounts for an unknown number of mixtures. That is, we rely on the (theoretical) latent-aware BIC as in Eq. (6), where the likelihood takes the form of Eq. (4).\
> To compute the BIC in practice, we need to consider  each $1\leq K\leq K_\max$ and use the EM algorithm to obtain estimates for the MLR parameters; finally, we then use the BIC score of the best such $K$ in our algorithm (and the corresponding estimates for the rest of the model parameters). We updated our manuscript text to make this explicit.
>
> We are happy to dispel any remaining doubts and clarify any persisting source of confusion.

---

> ### Comment · Reviewer_qK58 · 2025-08-06
>
> I thank the authors for their response. I would like to maintain my current evaluation of the paper. I have two minor comments:
> 1. **Regarding Question 1**: I recommend that the authors separate the bias term from the vector product in $f$—for example, by introducing an explicit bias term (e.g., $\beta_{j,k}^{(0)}$) instead of setting $\mathbf{Y}=1$.
>
>     My understanding is that $\mathbf{Pa}_j $ is a subset of $\mathbf{X}$, and for $X_1$, we have $\mathbf{Y}=\mathbf{Pa}_1=\emptyset$. Please correct me if I am wrong.
> 2. **Regarding Question 3(a)**: I suggest replacing $\mathbf{Y}$ with $\mathbf{Pa}$, or explicitly referring to Equation (3) or (4) to avoid confusion.

---

> > ### Author Response · Authors · 2025-08-07
> >
> > Thank you very much for your response and recommendations.
> >
> > Regarding Question 1, yes, your understanding is correct. We are happy to follow your suggestion to make bias term explicit and, regarding Question 3a, to make the reference to the equations explicit to avoid confusion.

---

### Official Review · Reviewer_yqWB · 2025-07-02

**Clarity:** 3
**Significance:** 3
**Originality:** 3
**Rating:** 5
**Confidence:** 3

**Summary:**

The authors consider the problem of performing causal discovery using observational datasets where samples come from multiple unknown distinct environments. They note that if samples come from heterogeneous subpopulations but this heterogeneity is ignored, applying traditional causal discovery methods may lead to spurious results. They observe that existing methods for addressing this setting, which use Gaussian Mixture Models (GMMs) to first split the dataset into estimated sub-populations, may lead to sub-optimal recovery of the subpopulations; in particular, GMMs may perform no-better than a random partition of the samples into subgroups. To address this issue, the authors propose Causal Mixture Modeling (CMM), which replaces the GMM-based subpopulation clustering with clustering based on a causal mixture of linear regressions (MLR).

They consider a setting where (observed) random variables X have distribution defined by a linear function of (unknown) causal relationships, with coefficients determined by (unobserved) latent categorical variables. Conditioned on its causal ancestors, this means that X is distributed following an (MLR) model. These distributions can be associated with a DAG containing nodes for both observed variables X and latent variables Z; edges between observed variables with causal relationships; and latent variables -> observed variables. Latent variables are assumed to have no incoming edges (no observed variables or other latent variables have causal effects on latent variables). The target problem is then to discover the causal subgraph on the set of observed random variables X, as well as the set of latent variables Z and the set of edges from Z to X.

One fundamental obstacle to solving this problem is that computing the Maximum Likelihood Estimate (MLE) of an MLR model is NP-hard. The authors first show that access to some oracle for computing the MLE of the MLR model for any number of mixtures, existing score-based frameworks for causal discovery (in particular Greedy Equivalent Search (GES)) can be extended to handle the latent-variable setup described above, and that the consistency results for GES hold under this extension.

To derive a practical algorithm motivated by the above observation, the authors propose using Expectation Maximization (EM) to approximate the MLE. They note that under certain conditions, the EM algorithm can find the MLE for MLR models if initialized appropriately. They replace the oracle rule with the EM algorithm, and then use the extended score-based framework developed based on GES to discover a topological ordering over the nodes of the DAG corresponding to X’s. They call this framework Discover a Causal Mixture Model (CMM).

**Questions:**

Is my understanding that the authors have not derived any guarantees on Algorithm 1 correct? If the components of the mixture are assumed to be well-separated, could the good-initialization regime for the EM algorithm described in lines 183-185 imply consistency for Algorithm 1, or would additional assumptions be necessary?

It is my understanding that the maximum number of latent variables Z identified by Algorithm 1 is capped at K_max, the user-specified maximum number of mixtures. Is this correct? Are there any heuristic rules that the authors would propose to help guide practitioners in selecting K_max and/or given multiple outputs from Algorithm 1 with different values of K_max, deciding which output is the most useful model of the data?

**Ethical Concerns:**

["NO or VERY MINOR ethics concerns only"]

**Final Justification:**

I maintain my score. In addition to the strengths I outlined in my initial review, I also endorse the paper because the authors discussed the main weakness I identified (the lack of consistency results for Algorithm 1) and provided a satisfactory discussion. I thus endorse this paper for acceptance.

**Limitations:**

yes

**Paper Formatting Concerns:**

Line 110: “mutatis mutandis” is a fun expression that I hadn’t heard before, and it seems to be used correctly based on my google search, but it is a bit obscure for some readers so perhaps consider replacing it with more conventional language (e.g., "modulo details") to improve accessibility.

**Quality:**

3

**Strengths And Weaknesses:**

Strengths:
The authors do a good job of motivating the problem of causal discovery with unknown heterogeneous subpopulations, and they clearly point out why existing methods (based on GMMs) do not adequately address this setting.
The MLR model studied and set of assumptions made seem pretty typical within the field of causal discovery, and none seem too unrealistic.

Weaknesses:
Though the oracle-MLE method describes inherits consistency results from GES, it is my impression that the authors do not establish any guarantees for Algorithm 1. If my understanding is incorrect, please correct me. However I consider this a fairly minor weakness, as the connection to the oracle-rule version is made very clear in the paper, and there are strong computational barriers to guaranteeing good approximations to the MLE so the authors’ framework is ultimately a reasonable approximation to a consistent method.

---

> ### Author Rebuttal · Authors · 2025-07-31
>
> We thank you for your response and appreciate your time in the review process.
>
> Please consider our response to your comments below.
>
>
> # Response to Questions
>  * **Consistency of Alg. 1.**\
>  Indeed, studying the consistency of the estimates derived by the EM algorithm is a difficult problem, and the literature is limited to rather strict settings (see the references in our manuscript). Therefore, it is correct that we cannot easily provide strict guarantees for Alg. 1. \
>     Having mentioned this, we can further note that the experimental evidence supports the connection we make in our conjecture. There is also a reasonable mechanism that could theoretically explain the observed behaviour (asymptotically).\
> 	Simply put, when the data truly follow the MLR model, it is more likely for our EM implementation to align with the oracle; when this does not happen, it is likely due to a faulty model, in which case our downstream process would anyways (asymptotically) reject the specific MLR hypothesis in favour of the correct alternative.\
> 	In summary, we confirm your observation that Alg. 1 is only guaranteed to be consistent under appropriate conditions, such as a good-enough EM-initialization regime, and should otherwise be viewed as a reasonable approximation to the problem.
>
> * **Choice of $K_\text{max}$.**\
> This is a fair point; indeed, we assume that a reasonably high maximum number of $K_\text{max}$ is given that caps the number of mixture components.\
> Here, the following comments may be made.
>
>    * First, even when $2 \leq K_\text{max}<K*$, for $K*$ the true number of clusters, the causal structure would still be (likely) correctly inferred, as the capped MLR would still offer a higher likelihood versus any alternative model (under appropriate assumptions, that is, non-zero additive noise and that the total back-door noise being higher than the remaining variance due to the mixture collapse.)
>    * Further, if the MLR model ever selects $K < K_\text{max}$, it is (asymptotically) sure (under the oracle assumption, assuming the latter to be capped, too) that this estimate of $K$ is the true number of mixture coefficients. When $K=K_\text{max}$, one could keep increasing the bound and repeat. Hence, we can consider this bound as more of an implementation issue than a truly limiting factor.
>    * For practitioners, we consider small values of $K_\text{max}$ (in the order of 10 or less) sufficient, with motivational examples in mind where $K$ counts distinct biological states, subpopulations, or conditions with distinct causal mechanisms, so that very large $K$ are most likely not meaningful. We recommend that users inspect the BIC values across different $K$ and increase the maximum if necessary.
>
>
> Feel free to let us know whether we sufficiently addressed the questions and if any concerns remain.

---

> > ### Comment · Reviewer_yqWB · 2025-08-03
> >
> > I thank the authors for their responses, which have addressed my questions. I will maintain my score. If accepted, I would suggest that the authors include some of their commentary on $K_{max}$ in the manuscript.

---

> > > ### Author Response · Authors · 2025-08-05
> > >
> > > Thank you for your response. We will include the above comments on $K_{max}$ in the revised manuscript.

---

### Decision · Program_Chairs · 2025-09-17

**Decision:**

Accept (poster)

**Comment:**

This work is about causal discovery using observational datasets where samples come from multiple unknown distinct environments. Two reviewers assigned clearly positive scores, highlighting the following strengths: good and convincing motivation of the problem, clear description of the approach, and convincing experiments.
In particular, one reviewer mentioned that the proposed extension of score-based causal discovery with the Bayesian Information Criterion has the potential to significantly enhance the practical applicability of statistical causal discovery methods.

There were also some critical concerns, mainly about (possibly too) strong assumptions, difficult interpretation of the inferred latent variables, and missing approximation guarantees. Most of these concerns, however, could be addressed by the authors in their rebuttal. I also found the rebuttal plausible and convincing, and for me, the positive aspects of this paper clearly outweigh the weaknesses. Therefore, I recommend acceptance.